# REPLAY MEMORY AS AN EMPIRICAL MDP: COMBINING CONSERVATIVE ESTIMATION WITH EXPERIENCE REPLAY

**Hongming Zhang**[1,2]*, **Chenjun Xiao**[1,2], **Han Wang**[1], **Jun Jin**[2], **Bo Xu**[3], **Martin Müller**[1]
[1]Department of Computing Science, University of Alberta
[2]Huawei Noah's Ark Lab
[3]Institute of Automation, Chinese Academy of Sciences
{hongmin2,chenjun,han8,mmueller}@ualberta.ca,
jun.jin1@huawei.com, boxu@ia.ac.cn

## ABSTRACT

Experience replay, which stores transitions in a replay memory for repeated use, plays an important role of improving sample efficiency in reinforcement learning. Existing techniques such as reweighted sampling, episodic learning and reverse sweep update process the information in the replay memory to make experience replay more efficient. In this work, we further exploit the information in the replay memory by treating it as an empirical *Replay Memory MDP (RM-MDP)*. By solving it with dynamic programming, we learn a conservative value estimate that *only* considers transitions observed in the replay memory. Both value and policy regularizers based on this conservative estimate are developed and integrated with model-free learning algorithms. We design the *memory density* metric to measure the quality of RM-MDP. Our empirical studies quantitatively find a strong correlation between performance improvement and memory density. Our method combines *Conservative Estimation with Experience Replay (CEER)*, improving sample efficiency by a large margin, especially when the memory density is high. Even when the memory density is low, such a conservative estimate can still help to avoid suicidal actions and thereby improve performance.

## 1 INTRODUCTION

Improving sample efficiency is an essential challenge for deep reinforcement learning (DRL). Experience replay (Lin, 1992) stores transitions in a replay memory and reuses them multiple times. This technique plays an important role for the success of DRL algorithms, such as Deep $Q$-Networks (DQN) (Mnih et al., 2015), Deep Deterministic Policy Gradient (DDPG) (Lillicrap et al., 2015; Haarnoja et al., 2018) and AlphaGo (Silver et al., 2016; 2017). DRL algorithms use gradient-based optimizers to incrementally update parameters. The learning requires several orders of magnitude more training samples than a human learning the same task. To speed up learning, many researchers focus on how to better process information in the replay memory before updating networks.

One research direction is to measure the relative importance of transitions and sample them with different priorities. Criteria for measuring the importance include temporal-difference (TD) error (Schaul et al., 2016), the "age" (Fedus et al., 2020) of transitions (Zhang & Sutton, 2017; Novati & Koumoutsakos, 2019; Sun et al., 2020; Wang et al., 2020; Sinha et al., 2022), errors of target values (Kumar et al., 2020a), and coverage of the sample space (Oh et al., 2021a; Pan et al., 2022). Another direction is to analyze the trait of transitions. For example, learning should start in a backward manner that allows sparse and delayed rewards to propagate faster (Lee et al., 2019); the best episodic experiences can be memorized and supervise the agent during learning (Lin et al., 2018); similar episodes may latch on and provide more information for update (Zhu et al., 2020; Hong et al., 2022; Jiang et al., 2022).

---

*Work done while an intern at Huawei Noah's Ark Lab.

Can we further improve the use of the replay memory? In our work, we consider the Replay Memory as an empirical MDP, called RM-MDP. Solving RM-MDP provides us an estimate that considers all existing transitions in the replay memory together, not just single samples or trajectories at a time. The estimate is conservative because some actions may not be contained in the memory at all. We propose to use this conservative estimate $\hat{Q}$ from RM-MDP to regularize the learning of the $Q$ network on the original MDP. We design two regularizers, a value regularizer and a policy regularizer. The value regularizer computes the target value by combining the estimates from $\hat{Q}$ and $Q$ network. For the policy regularizer, we derive Boltzmann policies from the $\hat{Q}$ and $Q$ network, and constrain the distance between the two policies using Kullback–Leibler (KL) divergence.

Our contribution is four-fold: (1) We consider the replay memory as an empirical Replay Memory MDP (RM-MDP) and obtain a conservative estimate by solving this empirical MDP. The MDP is non-stationary and is updated efficiently by sampling. (2) We design value and policy regularizers based on the **C**onservative **E**stimate, and combine them with **E**xperience **R**eplay (CEER) to regularize the learning of DQN. (3) We introduce *memory density* as a measure of the quality of RM-MDP, and empirically find a strong correlation between the performance improvement and memory density. The relationship gives us a clear indication of when our method will help. (4) Experiments on Sokoban (Schrader, 2018), MiniGrid (Chevalier-Boisvert et al., 2018) and MinAtar (Young & Tian, 2019) environments show that our method improves sample efficiency by a large margin especially when the memory density is high. We also show that even if the memory density is low, the conservative estimate can help to avoid suicidal actions and still benefit the learning, and our method is effective in environments with sparse and delayed rewards.

## 2 RELATED WORK

**Reweighted Experience Replay.** Since DQN (Mnih et al., 2015) replays experiences uniformly to update $Q$ network and achieves human-level performance on Atari games, improving experience replay (Lin, 1992) became an active direction. Prioritized Experience Replay (PER) (Schaul et al., 2016) is the first and most famous improvement. It replays important transitions as measured by TD errors more frequently. Oh et al. (2021b) additionally learn an environment model and prioritize the experiences with both high model estimation errors and TD errors. Saglam et al. (2022) show that for actor-critic algorithms in continuous control, actor networks should be trained with low TD error transitions. Sampling recent transitions is another popular metric (Novati & Koumoutsakos, 2019; Sun et al., 2020; Wang et al., 2020; Cicek et al., 2021). These authors argue that old transitions will hurt the update since the current policy is much different from previous ones. Distribution Correction (DisCor) (Kumar et al., 2020a) reweights transitions inversely proportional to the estimated errors in target values. For these rule-based sampling strategies, Fujimoto et al. (2020) show that correcting the loss function is equivalent to non-uniform sampling, which provides a new direction for experience replay. Besides, learning a replay policy (Zha et al., 2019; Oh et al., 2021a) is also a feasible approach. These methods alternately learn a replay policy and a target policy.

**Experience Replay with Episodic RL.** Episodic RL (Blundell et al., 2016; Pritzel et al., 2017; Hansen et al., 2018) uses a separate lookup table to memorize the best experiences ever encountered. Episodic Memory Deep $Q$-Networks (EMDQN) (Lin et al., 2018) uses the maximum return from episodic memory as target value. This supervised learning compensates for the slow learning resulting from single step reward updates. Episodic Reinforcement Learning with Associative Memory (ERLAM) (Zhu et al., 2020) builds a graph on top of state transitions and uses it as early guidance. The RM-MDP construction in our work is somewhat similar, but more general. We construct the MDP and solve it through a modified dynamic programming scheme. We have no constraint on the MDP, so it can be stochastic and can have cycles.

**Experience Replay with Reverse Update.** The work by Dai & Hansen (2007); Goyal et al. (2019) focuses on the sequential character of trajectories. They argue that a state's correct $Q$ value is preconditioned on the accurate successor states' $Q$ values. Episodic Backward Update (EBU) (Lee et al., 2019) samples a whole episode and successively propagates the value of a state to its previous states. Topological Experience Replay (TER) (Hong et al., 2022) builds a graph to remember all the predecessors of each state, and performs updates from terminal states by successively moving backward via breadth-first search. Our method propagates rewards when solving RM-MDP, thereby the sequential character of trajectories is considered in the regularizers.

**Model-based RL.** A replay memory can be considered as a non-parametric model ([Hasselt et al., 2019](#)). The main difference is that the replay memory only contains true experience, while a model can generate imaginary transitions. Dyna ([Sutton, 1991](#); [Sutton et al., 2012](#)) uses a model to generate imaginary transitions and updates the agent with both real and imaginary transitions. Model-based value expansion (MVE) ([Feinberg et al., 2018](#); [Buckman et al., 2018](#); [Xiao et al., 2019](#)) uses a model to simulate on-policy rollouts and provides higher-quality target values. The AlphaGo and AlphaZero algorithms alternately optimize a reactive (neural network) and a non-reactive (tree search) policy ([Silver et al., 2016](#); [2017](#); [2018](#); [Sun et al., 2018](#)). World Models ([Ha & Schmidhuber, 2018](#)) and Dreamer ([Hafner et al., 2020](#); [Hafner et al., 2021](#)) focus on how to learn a high-quality model. Model-based RL has a fundamental appeal since it can generate imaginary transitions and improve sample efficiency. The challenges are that it is hard to learn a good model, and the wall-clock training time increases dramatically due to model learning and model planning.

## 3 BACKGROUND

**Markov Decision Process (MDP).** The interaction of an agent with its environment can be modeled as a MDP $\mathcal{M} = (\mathcal{S}, \mathcal{A}, R, P, \rho_0, \gamma)$. $\mathcal{S}$ is the state space, $\mathcal{A}$ is the action space, $P : \mathcal{S} \times \mathcal{A} \times \mathcal{S} \rightarrow [0, 1]$ is the environment transition dynamics, $R : \mathcal{S} \times \mathcal{A} \times \mathcal{S} \rightarrow \mathbb{R}$ is the reward function, $\rho_0 : \mathcal{S} \rightarrow \mathbb{R}$ is the initial state distribution and $\gamma \in (0, 1)$ is the discount factor. The goal of the agent is to learn an optimal policy that maximizes the expected discounted sum of rewards $\mathbb{E}[\sum_{t=0}^{\infty} \gamma^t r_t]$. $Q$-learning is a classic algorithm to learn the optimal policy. It learns the $Q$ function with Bellman optimality equation ([Bellman, 2021](#)), $Q^*(s, a) = \mathbb{E}[r + \gamma \max_{a'} Q^*(s', a')]$. An optimal policy is then derived by taking an action with maximum $Q$ value at each state. DQN ([Mnih et al., 2015](#)) scales up from tabular $Q$-learning by using deep neural networks and experience replay ([Lin, 1992](#)).

**Solving An Empirical MDP**. We treat the replay memory $\hat{M}$ as an empirical *Replay Memory MDP (RM-MDP)*, $\hat{\mathcal{M}} = (\mathcal{S}, \mathcal{A}, R, \hat{P}, \rho_0, \gamma)$. All components are the same as $\mathcal{M}$ except $\hat{P}$. $\hat{P}$ is the empirical transition dynamics summarized from the transitions in $\hat{M}$ collected by behavior policy $\beta$. We set $\hat{p}(s, a, s') = 0$ if $\beta(a|s) = 0$. Let $\hat{Q}$ be the action value function of the RM-MDP. Solving RM-MDP equals to solving the following equation:

$$\hat{Q}^*(s, a) = \mathbb{E}_{s' \sim \hat{P}(s'|s, a)}[r + \gamma \max_{a' : \beta(a'|s') > 0} \hat{Q}^*(s', a')], \tag{1}$$

where $\hat{Q}^*$ is the optimal action value function for RM-MDP[1]. This is a fundamental problem in offline/batch reinforcement learning ([Lange et al., 2012](#)). The learning goal is to maximize the cumulative reward limited to a static dataset. This returns a conservative policy with best possible performance. The constraint avoids distributional shift caused by taking actions outside of the dataset when estimating target values. Though the replay memory in online learning is usually a time-changing first in first out queue, at a specific time step, the memory can be considered as a static dataset that contains finite number of transitions. Thus, solving RM-MDP is similar to offline RL setting that solves an empirical MDP and results in a conservative estimate ([Levine et al., 2020](#)).

## 4 METHOD

We regularize online RL agents with the conservative estimation from replay memory. Figure [1](#) shows the overview of our method. We first construct the RM-MDP with transitions in the replay memory. We then get a conservative estimate by solving RM-MDP. Finally, we design value and policy regularizers based on the estimate to help online learning. In Section [4.1](#), we describe the construction of the RM-MDP by building the replay memory as a graph. In Section [4.2](#), we describe solving the RM-MDP. In Section [4.3](#), we describe how we design the value and policy regularizers and summarize the whole method.

### 4.1 BUILDING RM-MDP

The replay memory changes overtime during online learning, and offline DRL methods usually take tens of thousands of gradient steps to extract a good policy ([Fujimoto et al., 2019](#); [Kumar et al.,](#)

---

[1]We use $Q$ and $\hat{Q}$ to denote the estimates from the network and RM-MDP respectively.

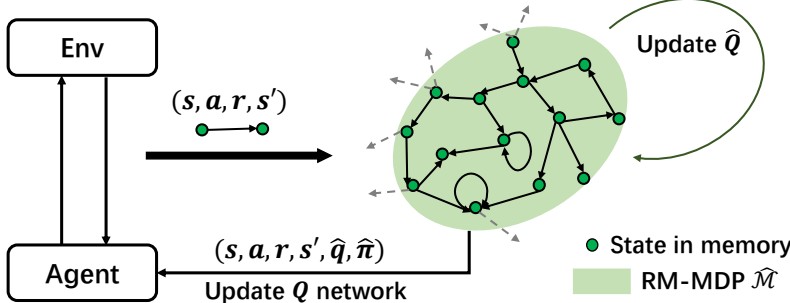

Figure 1: An overview of our method. Transitions from the interaction with environment constructs RM-MDP, which is part of the environment MDP. Green circles denote states in the memory. The solid lines denote existing transitions, and dashed lines denote transitions not in the memory. We obtain value and policy estimates from RM-MDP, which will be provided to regularize the learning.

2020b; Kostrikov et al., 2022). To derive a quick and stable estimation from RM-MDP, we focus on constructing the replay memory as a graph, which can be solved with tabular methods.

**Data structure.** As introduced in Section 3, $\hat{M}$ is the replay memory and $\hat{\mathcal{M}} = (\mathcal{S}, \mathcal{A}, R, \hat{P}, \gamma)$ is the corresponding RM-MDP. To obtain $\hat{\mathcal{M}}$, we construct the replay memory as a directed graph $\mathcal{G}$:

$$\mathcal{G} = (V, E), V = \{\phi(s)|(s, \hat{Q}(s, \cdot))\}, E = \{\phi(s) \xrightarrow{a} \phi(s')|(a, r, N(s, a, s'))\}. \quad (2)$$

In the graph $\mathcal{G}$, $V$ denotes the set of vertices, $E$ denotes the set of edges. $V$ contains states $s$ and action value estimation $\hat{Q}(s, \cdot)$. $\phi$ is an embedding function to map the high-dimensional states to a low-dimensional representation for fast query (keys in the graph $\mathcal{G}$). We store vertices in a dictionary with $\phi$ as a hashing function serving as keys, thus we can retrieve a state in $\mathcal{O}(1)$ time. $\hat{Q}(s, \cdot)$ only contains values for actions that were taken by behavior policy $\beta$, i.e., $a : \beta(a|s) > 0$. For each edge $e \in E$, we store actions $a$, rewards $r$ and visit counts $N(s, a, s')$. These statistics are similar to those in AlphaGo (Silver et al., 2016; 2017; 2018) , which are stored in a tree structure and used for Monte Carlo Tree Search (MCTS) (Kocsis & Szepesvári, 2006; Browne et al., 2012). Each edge is identified by the tuple $(\phi(s), a, \phi(s'))$, thus we can also query the statistics on an edge in $\mathcal{O}(1)$ time. Our graph has no limitation on the structure of the RM-MDP. The graph can have cycles and the environment dynamics can be stochastic. We estimate $\hat{P}$ for transition $(s, a, s')$ with visit counts:

$$\hat{p}(s'|s, a) = \frac{N(s, a, s')}{\sum_{s'} N(s, a, s')}. \quad (3)$$

If a state or an action has never been tried $N(s, a) = 0$, we define $\hat{p}(s'|s, a) = 0$ for all $s' \in \mathcal{S}$.

**Update rule.** Besides the graph structure, the memory also maintains the same features as the replay memory used in DQN. It stores the most recent experiences with a fixed memory size. Since all these statistics are scalars, the memory consumption is mostly on account of states $s$ in vertices $V$. We count the memory size by the number of vertices. When a new transition $(s, a, r, s')$ comes, we first check if $\phi(s)$ exists in the graph. If not, we create a new vertex and edge and initialize $\hat{Q}(s, a) = 0, N(s, a, s') = 1$. Otherwise, if $\phi(s)$ already exists, we merge the same states into one node and update the visit count $N(s, a, s') = N(s, a, s') + 1$. Different trajectories that have the same state representation will intersect. If there is no intersection between trajectories, the graph degenerates to separate connected components equivalent to the replay memory used in DQN.

### 4.2 Solving RM-MDP

With the graph $\mathcal{G}$, we have all elements of RM-MDP $\hat{\mathcal{M}}$. We use value iteration to update $\hat{Q}$:

$$\hat{Q}(s, a) \leftarrow \sum_{s'} \hat{p}(s'|s, a)[r + \gamma \max_{a':\beta(a'|s')>0} \hat{Q}(s', a')], \quad (4)$$

where $\hat{p}$ is the empirical dynamics computed by equation 3, $\gamma$ is the discount factor, $\hat{Q}(s', \cdot)$ and $r$ can be queried from corresponding vertex and edge in $\mathcal{O}(1)$ time. During the learning, each environment interaction step will add a new transition to the memory, which makes RM-MDP non-stationary. Instead of updating all transitions of the new RM-MDP, we only perform the update (4)

on a sampled batch of transitions. In DQN, the $Q$ function is updated with a sampling batch every four environmental steps (Mnih et al., 2015). We use the same batch to update $\hat{Q}$. Besides, when an episode ends, we update $\hat{Q}$ along the episode in reversed order. This is similar to EBU (Lee et al., 2019) but we do the backward update for the memory rather than for $Q$ networks. The sampling-based update and episode-based update allow sparse and delayed rewards to propagate quickly through all transitions of related trajectories. There is no extra computation for sampling, and the time consumption for equation 4 is minor compared to the update cost of $Q$ networks.

### 4.3   CEER: Regularizing Deep Q Networks by RM-MDP

With the conservative estimation from replay memory, we design value and policy regularizers.

**Value regularizer.** DQN is known to suffer from substantial overestimation in some games due to the combination of deep neural networks with the maximization step in Bellman optimality equation (Van Hasselt et al., 2016). Since $\hat{Q}$ from the RM-MDP is a conservative estimation and is estimated in a tabular manner without function approximation, it is not so prone to overestimation. It can be a good complement to compute a better target $Q$ value. Let $q(s, a) = r + \gamma \max_a Q(s', a)$ be the target $Q$ value estimated by $Q$ network for state-action pair $(s, a)$, and $\hat{q}(s, a)$ computed by equation 4 be the value estimated from RM-MDP. The target value for state action pair $(s, a)$ is:

$$q_{\text{target}}(s, a) = (1 - \alpha)q(s, a) + \alpha\hat{q}(s, a), \tag{5}$$

where $\alpha \in [0, 1]$ is a parameter. Using mean square error with a sample batch $\mathcal{D}$, our TD loss is:

$$\mathcal{L}_{TD} = \frac{1}{|\mathcal{D}|} \sum_{(s,a) \in \mathcal{D}} (q_{\text{target}}(s, a) - Q(s, a))^2. \tag{6}$$

**Policy regularizer.** Since the purpose of learning is to derive an optimal policy from the $Q$ network, we design a policy regularizer to accelerate policy learning. Let $\hat{\pi}(s) = \text{softmax}_{\tau, a:\beta(a|s)>0}(\hat{Q}(s, \cdot))$ be the Boltzmann policy derived from $\hat{Q}$ at state $s$, where $\tau$ is the softmax temperature. Action $a$ is constrained by behavior policy $\beta$, thus policy $\hat{\pi}$ is conservative. Similarly, policy $\pi(s) = \text{softmax}_{\tau, a:\beta(a|s)>0}(Q(s, \cdot))$ is derived from the $Q$ network. We regularize policy $\pi$ by minimizing the Kullback–Leibler (KL) divergence between $\hat{\pi}$ and $\pi$:

$$\mathcal{L}_{policy} = \frac{\lambda}{|\mathcal{D}|} \sum_{s \in \mathcal{D}} D_{KL}(\hat{\pi}(s) \parallel \pi(s)), \tag{7}$$

where $\lambda > 0$ is a parameter, and $\mathcal{D}$ is a batch of experiences. The intuition for this policy regularizer is that because policy $\hat{\pi}$ is derived from a conservative estimation, it lower bounds the performance of policy $\pi$. If the RM-MDP $\hat{\mathcal{M}}$ is a good approximation of a local part of the environment MDP $\mathcal{M}$, then $\hat{\pi}$ approximates the optimal policy in $\mathcal{M}$ and thereby accelerate the learning. Even if the approximation is not so good, the conservative $\hat{\pi}$ can still help by avoiding suicidal actions in some games and thereby improves the performance. Combining the two regularizers, the update loss is:

$$\mathcal{L} = \mathcal{L}_{TD} + \mathcal{L}_{policy}. \tag{8}$$

The pseudo-code for CEER is shown in Algorithm 1.

## 5   Experiments

In this section, we aim to answer the following questions: (1) Does conservative estimation-based experience replay (CEER) speed up learning? (2) In which circumstances will CEER help? (3) What are the effects of value and policy regularizers? How does CEER compare to learning from $\hat{Q}$ in a supervised manner similar to EMDQN (Lin et al., 2018) and ERLAM (Zhu et al., 2022)?

### 5.1   Setup

**Environments.** We evaluate our method on Sokoban (Schrader, 2018), MiniGrid (Chevalier-Boisvert et al., 2018) and MinAtar (Young & Tian, 2019) based on OpenAI Gym interface (Brockman et al., 2016). Sokoban is Japanese for warehouse keeper and a traditional puzzle. The player

---

**Algorithm 1** Combine Conservative Estimation with Experience Replay (CEER)

---
1: Initialize replay memory $\hat{M}$ with fixed memory size
2: Initialize action value network $Q$ with weights $\theta$, target network $\bar{Q}$ with weights $\bar{\theta} \leftarrow \theta$
3: Initialize the environment $s_0 \leftarrow Env$
4: **for** environments step $t = 0$ **to** $T$ **do**
5:      Select an action $a_t$ by $\epsilon$-greedy with respect to $Q(s_t, \cdot; \theta)$
6:      Execute $a_t$ in $Env$ and get $r_t, s_{t+1}$
7:      Store the transition $(s_t, a_t, r_t, s_{t+1})$ in $\hat{M}$ and update visit count $N(s_t, a_t, s_{t+1})$
8:      Sample random minibatch of transitions from $\hat{M}$
9:      Update $\hat{Q}(s_t, a_t)$ via equation 4 and store the result to $\hat{M}$
10:     Update $Q(s_t, a_t; \theta)$ by minimizing $\mathcal{L}$ via equation 8
11:     Update $\bar{\theta} \leftarrow \theta$ every $C$ steps
12: **end for**

---

has to push all boxes in a room to the storage locations without getting blocked. MiniGrid implements many tasks in the grid world environment. Most of the games are designed with sparse rewards. MinAtar is an image-based miniaturized version of Atari games (Bellemare et al., 2013) which maintains the mechanics of the original games as much as possible and is much faster. For Sokoban and MiniGrid, the map is randomly generated at each episode. For MinAtar, objects are randomly generated at different time steps. So all these environments require the policy to generalize across different configurations. More details about these environments can be found in Appendix B.

**Baselines and Implementation Details.** We compare our method with DQN (Mnih et al., 2015), PER (Schaul et al., 2016), EBU (Lee et al., 2019), DisCor (Kumar et al., 2020a) and TER (Hong et al., 2022). We use the same network for all algorithms. The architecture is the same as in the MinAtar baselines. The network contains a convolutional layer with $16$ $3 \times 3$ convolutions with stride 1, and a fully connected hidden layer with 128 units. We search the learning rates for baselines among {3e-3,1e-3,3e-4,1e-4,3e-5,1e-5} and report the best performance. For our method, we fix the learning rate to 1e-4 and search our particular parameters. Value regularizer weight $\alpha \in \{0,0.2,0.5,0.8,1\}$ and policy regularizer weight $\lambda \in \{0,0.01,1,2,5\}$. All other common parameters are set the same as shown in Appendix A Table 2. We run each experiment with 20 different random seeds and show the mean score and the standard error with solid line and shaded area. Each run consists of 2 million steps of interaction with the environment. The performance is evaluated by running 100 episodes after every 100k environmental steps. More details can be found in Appendix A.

## 5.2 A TOY EXAMPLE

We first give a toy example to show the benefit of regularizing DQN with conservative estimation. We consider Cliffworld (Sutton & Barto, 2018) as shown in Figure 5 Appendix B. There are 48 states and 4 actions. The goal is to reach state G starting from state S. The reward of reaching G is 1, dropping to the cliff gives -1, otherwise is 0. This case can be solved by a tabular method such as $Q$-learning easily, but with networks such as DQN, learning is more difficult because of incremental update and parameter interference.

We show the learning curves of mean score, and the mean square error of the $Q$ value in Figure 2. **Only_value** and

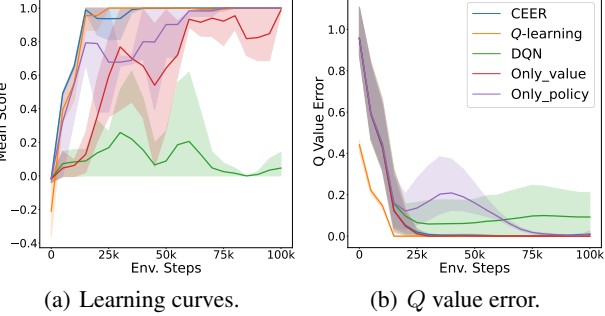

(a) Learning curves.     (b) $Q$ value error.

Figure 2: **(a)** The learning curve of each algorithm. CEER achieves comparable learning speed with $Q$-learning. The policy regularizer contributes more on policy learning. **(b)** The value error compared with optimal $Q$ value during learning. The value regularizer contributes more on reducing the value error.

**Only_policy** denote the methods with only one regularizer. CEER uses both regularizers. DQN uses no regularizer and $Q$-learning is a tabular method without a network. DQN struggles and cannot find

the goal state within 100k interactions. With the two regularizers, the performance improves a lot and the learning speed is on par with $Q$-learning. The performance of **Only_value** and **Only_policy** demonstrate that each regularizer plays an important and different role. The value regularizer helps learn the optimal value faster. The policy regularizer helps the policy learning, while the $Q$ value loss decreases slower. In this simple case, the replay memory contains almost all the transitions so that we can consider $\hat{\mathcal{M}} \approx \mathcal{M}$, i.e., the RM-MDP is almost the same as the environment MDP, and the conservative policy is a perfect approximation of the optimal policy for $\mathcal{M}$. Next, we show that the two regularizers still help in more complex environments, even if $\hat{\mathcal{M}}$ is much smaller than $\mathcal{M}$.

## 5.3 OVERALL PERFORMANCE

We show the final scores on Sokoban, MiniGrid and MinAtar in Table 1. Our method achieves the best performance in most of the environments, especially in hard environments with sparse and delayed rewards. We evaluate our method on seven Sokoban environments with increasing difficulty. For example, Push-5×5-1 is a problem with a 5×5 map and 1 box to push. In easy environments such as Push-5×5-1, almost all methods achieve the best performance. When difficulty increases, only our method can handle these environments such as Push-7×7-2. Similar results can be found in MiniGrid. When the difficulty increases, our method outperforms the baselines by a large margin, for example in LavaCrossing-Hard. These results show that CEER is good at handling environments with sparse and delayed rewards. Besides the above goal-conditioned and navigation tasks, our method can also handle video games with complex mechanics. In MinAtar, CEER performs best or achieves competitive performance in four out of five games. Especially on Breakout, the final score is double that of other algorithms. PER samples transitions with high TD error more frequently. However, the TD error changes dynamically as the network is updated, which may hurt the performance (Cicek et al., 2021). EBU and TER consider the sequential character of trajectories and sample transitions in reversed order. Our method propagates rewards when solving RM-MDP, thereby also considers the character of trajectories. Furthermore, our regularizers update the network in a supervised manner, which is more stable than TD learning. DisCor samples transitions proportional to the accuracy of the target value, but such transitions are not necessarily the most related to the optimal policy. In contrast, our method aims to provide accurate target values by combining the conservative estimation $\hat{Q}$ from RM-MDP with network estimation $Q$, and treats all transitions as equally important. This can provide accurate target values and does not miss the most related transitions. In summary, our method combines the advantages of several previous methods, and our experimental results indicate that it is suitable for a large variety of different tasks. Besides the final scores, we show the learning curves for each environment in Appendix C Figures 9, 10, and 11.

Table 1: Overall performance on Sokoban, MiniGrid and MinAtar environments. Numbers in bold represent the method that achieves the highest final scores.

| Environment | | DQN | DDQN | PER | EBU | DisCor | TER | CEER |
|---|---|---|---|---|---|---|---|---|
| | Push-5×5-1 | **10.82** | 10.78 | 10.81 | 7.68 | 10.76 | **10.82** | **10.82** |
| | Push-5×5-2 | 10.52 | 8.72 | 9.61 | 6.12 | 8.39 | 11.10 | **11.62** |
| | Push-6×6-1 | 9.26 | 0.01 | 0.33 | -3.55 | -0.04 | 8.48 | **10.73** |
| Sokoban | Push-6×6-2 | -1.93 | -9.71 | -9.81 | -11.59 | -10.43 | -10.40 | **10.54** |
| | Push-6×6-3 | -10.06 | -10.03 | -10.27 | -11.71 | -10.56 | -11.80 | **7.65** |
| | Push-7×7-1 | 5.78 | -5.60 | -4.61 | -6.39 | -4.14 | -0.69 | **10.65** |
| | Push-7×7-2 | -2.96 | -11.76 | -11.60 | -11.86 | -11.81 | -11.90 | **9.97** |
| | DoorKey | 1.19 | 3.43 | 3.45 | -3.60 | -0.63 | -3.60 | **3.50** |
| | Unlock | -0.87 | 2.72 | 2.74 | -2.88 | -2.72 | 2.59 | **2.80** |
| | RedBlueDoors | 5.40 | 4.67 | 7.06 | -7.22 | 5.77 | 6.70 | **7.09** |
| MiniGrid | SimpleCrossing-Easy | 3.08 | 3.09 | 3.07 | 2.98 | 3.02 | 3.08 | **3.11** |
| | SimpleCrossing-Hard | 3.03 | 3.02 | 3.02 | -2.85 | 2.70 | 3.01 | **3.09** |
| | LavaCrossing-Easy | 0.58 | 2.24 | 1.52 | -3.00 | -1.13 | -2.28 | **3.10** |
| | LavaCrossing-Hard | -3.34 | -3.37 | -3.38 | -3.31 | -3.38 | -3.09 | **3.05** |
| | Asterix | 19.82 | 20.1 | 25.8 | 26.0 | **31.19** | 3.07 | 19.82 |
| | Breakout | 13.57 | 16.25 | 17.76 | 19.97 | 17.14 | 12.73 | **42.53** |
| MinAtar | Freeway | 57.85 | 57.66 | 57.79 | **59.07** | 56.85 | 22.08 | 58.77 |
| | Seaquest | 14.73 | **18.57** | 17.48 | 12.65 | 16.69 | 1.62 | 17.96 |
| | SpaceInvaders | 50.71 | 51.15 | 55.09 | 71.83 | 57.85 | 45.29 | **73.43** |

## 5.4 RELATIONSHIP OF PERFORMANCE IMPROVEMENT AND CONSERVATIVE ESTIMATION

To better understand the relationship between performance improvement and conservative estimation, we design *Average Relative Performance Improvement* (ARPI) to measure the improvement for the whole learning process, and *Memory Density* to measure the quality of RM-MDP.

**Average Relative Performance Improvement (ARPI).** Suppose we want to compare the performance of method $A$ with method $B$. Let $R_k(A)$ and $R_k(B)$ be the cumulative rewards of $A$ and $B$ at evaluating iteration $k$. $R_{\max} = \max(\max_k R_k(A), \max_k R_k(B))$ is the maximum cumulative reward that the two methods achieved during the learning process. We define ARPI($A|B$) as:

$$\text{ARPI}(A|B) = \frac{1}{K}\sum_{k=1}^{K}\frac{R_k(A) - R_k(B)}{R_{\max} - R_k(B)} \le 1. \tag{9}$$

ARPI($A|B$) presents the relative performance difference of method $A$ compared with method $B$. If ARPI($A|B$) $> 0$, it means $A$ performs better than $B$. When $A$ achieves the maximum cumulative reward $R_{\max}$ at each evaluation $k$, ARPI($A|B$) $= 1$ shows the maximum performance improvement.

**Memory Density.** We measure the quality of estimation from RM-MDP based on the density of memory. The intuition here is, if we have tried more different actions at a state, we should have a better understanding of the state. Let $D_{\text{out}}(s)$ be the out-degree of the vertex at state $s$, $|\mathcal{A}|$ the size of action space, $\hat{M}$ the replay memory. We define the average out-degree of the memory as $D_{\text{out}}(\hat{M}) = \frac{1}{|\hat{M}|}\sum_{s\in\hat{M}} D_{\text{out}}(s)$. Since states are collected as trajectories, usually there is at least one out-degree for each state vertex. The maximum out-degree is the size of action space, so $1 \le D_{\text{out}}(\hat{M}) \le |\mathcal{A}|$. We assume $|\mathcal{A}| > 1$ and define the memory density as:

$$\text{Density}(\hat{M}) = \frac{D_{\text{out}}(\hat{M}) - 1}{|\mathcal{A}| - 1} \in [0, 1]. \tag{10}$$

We subtract 1 in both numerator and denominator to normalize Density($\hat{M}$) between 0 and 1.

**Correlation between ARPI and Density.** We compare CEER with DQN algorithm and compute the Pearson Correlation Coefficient $\rho$ (PCC) (Pearson, 1896; Lee Rodgers & Nicewander, 1988) to quantify this correlation. PCC is a measure of linear correlation between two sets of data. We measure a very strong linear correlation $\rho = 0.881$ between performance improvement and memory density. In Figure 3, each dark point denotes an environment, and the line is the best linear regression. When the memory density increases, the performance improves. Even with a low memory density, on MinAtar environments Asterix and SpaceInvaders, there are still improvements. The conservative estimation can help the agent avoid suicidal actions, thereby the agent lives longer and gets higher reward. One exception is Freeway, though the final performance is competitive, the whole learning is slower compared with DQN as shown

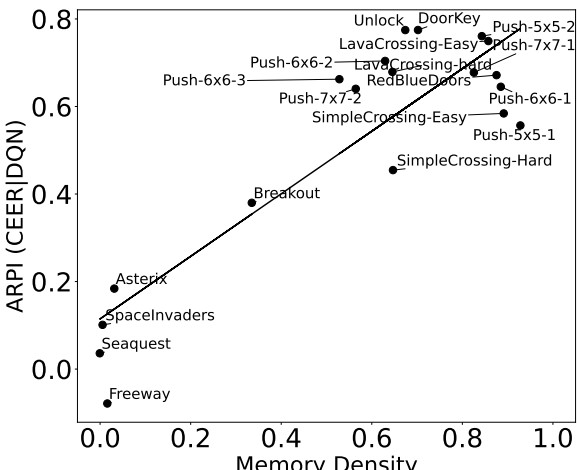

Figure 3: Relationship between memory density and the performance improvement. We compare CEER with DQN. The memory density is the average value during learning. The oblique line shows the result of linear regression. There is a strong correlation between memory density and ARPI.

in Appendix C Figure 11. In this environment, the randomly generated objects will move with different speeds as the game progresses. This feature is not included in the state representation, which breaches the Markov property of MDP. The graph merges different states that look the same but have different speeds, which leads to wrong estimation.

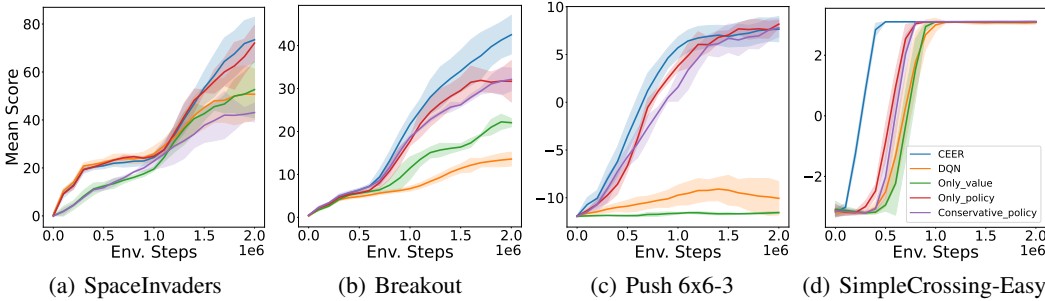

(a) SpaceInvaders    (b) Breakout    (c) Push 6x6-3    (d) SimpleCrossing-Easy

Figure 4: Ablation studies of the two regularizers across environments with different memory densities. **Only_value** and **Only_policy** only use one regularizer, **Conservative_policy** is the policy derived from RM-MDP by supervised learning. CEER combines both regularizers.

## 5.5 ABLATION STUDY

We study two questions: (1) What are the effects of the value and policy regularizers on environments with different memory density? (2) What is the performance of the conservative policy derived from the replay memory? We choose four environments with increasing memory densities, SpaceInvaders (0.01), Breakout (0.34), Sokoban 6×6-3 (0.53), and SimpleCrossing-Easy (0.89).

Figure 4 compares the learning curves that with 0, 1 and 2 regularizers. Both regularizers independently benefit the learning. The policy regularizer plays a more important role on performance improvement. On Sokoban Push-6×6-3, **Only_value** performs worse than DQN, since though the mean square error is reduced, the correct order of actions is not maintained due to the function approximation error induced by the neural network. This is similar to the result on Cliffworld in Figure 2, the $Q$ value error of **Only_value** reduces faster than **Only_policy** while the performance improves slower. The curves of **Conservative_policy** show results of the conservative policy derived from the replay memory. **Conservative_policy** remove the target estimation from $Q$ network, so it learns from $\hat{Q}$ in a supervised manner with value weight $\alpha = 1$. This is similar to EMDQN (Lin et al., 2018) and ERLAM (Zhu et al., 2020). The difference is that we still maintain the policy regularizer since we find with only supervised learning from $\hat{Q}$, the correct order of actions is hard to maintain and results in worse performance. Figure 4 shows that when memory density is high, **Conservative_policy** outperforms DQN. The conservative estimation from RM-MDP accelerates the learning in these scenarios. However, it is worse than CEER, which emphasizes the importance of combining estimation from both RM-MDP and $Q$ network in target value. When the memory density decreases, **Conservative_policy** performs worse than DQN. This could be because the conservative policy discourages exploration and thereby slows down learning. This indicates that though the conservative estimation can avoid suicidal actions, it will also miss optimal actions. In this case, bootstrapping from $Q$ network works better.

## 6 CONCLUSION

We propose to consider the replay memory as an empirical Replay Memory MDP (RM-MDP). Solving this RM-MDP combines existing experiences in the replay memory into a conservative estimate. We design two regularizers to accelerate online DRL. We empirically show the effectiveness of our method in many environments and observe a strong correlation between the performance improvement and memory density of RM-MDP. Results show that our method improves the sample efficiency by a large margin, especially when the memory density is high. Even with a low memory density, a conservative estimate can help to avoid suicidal actions and thereby improve performance. Our method is also suitable for environments with sparse and delayed rewards.

There are two main directions for future work. Since we have shown a strong correlation between memory density and performance improvement, designing state abstraction methods to merge information from similar states is a significant direction, which can increase the memory density and further improve the performance. Another challenge is extending CEER to environments with continuous action space. We can use discretization or clustering to construct a discrete RM-MDP with a graph structure, or solve it using existing sampling-based offline RL methods.

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

APPENDIX

## A    IMPLEMENTATION DETAILS

**Network Architecture.** For the toy example Cliffworld (Sutton & Barto, 2018) in Section 5.2, we use a network with two fully connected layers. Each layer has 64 units, and we use tanh as activation function. $Q$-learning does not use network, it maintains a $Q$ table with size $|\mathcal{S}| \times |\mathcal{A}|$.

For Sokoban, MiniGrid and MinAtar environments, We use a convolutional neural network. The architecture is the same as used in MinAtar (Young & Tian, 2019) baselines. It consists of a convolutional layer, followed by a fully connected layer. The convolutional layer has 16 $3 \times 3$ convolutions with stride 1, the fully connected layer has 128 units. These settings are one quarter of the final convolutional layer and fully connected layer of the network used in DQN (Mnih et al., 2015).

**Hyper-parameters.** Our method is based on DQN, other baselines additionally incorporate double $Q$-learning technique (Van Hasselt et al., 2016). We maintain most parameters the same as original DQN and reduce the interaction steps to run more different random seeds. We run each experiment with 2 million steps of interaction with the environment. We proportionally reduce other parameters based on the interaction steps. The $\epsilon$-greedy exploration is linearly decayed from 1 to 0.01 in 1 million steps. The target network is updated every 1000 steps. The replay memory size is set as 100,000. The minibatch size is 32. The replay ratio is 0.25 (Fedus et al., 2020), that is the $Q$ function is updated once per four environmental steps. The optimizer for the network is Adam. The discount factor is 0.99. Table 2 shows the details of hyper-parameters that used for all methods.

Besides the common parameters, there are other parameters that are specific to different methods. For PER (Schaul et al., 2016), the parameter $\alpha$ in sampling probability is set as 0.4, and the parameter $\beta$ in importance-sampling (IS) weights is linearly annealed from 0.4 to 1 within 0.5 million environmental steps. For EBU (Lee et al., 2019), the diffusion factor $\beta$ is set as 0.5. For DisCor (Kumar et al., 2020a), the initial temperature $\tau$ is set as 10, and the error network $\Delta_\theta$ has one extra fully connected layer as suggested in original paper. For TER (Hong et al., 2022), the batch mixing ratio $\eta$ is 0.5 for MiniGrid and MinAtar, and 0.1 for Sokoban. We search the replay memory size for TER within {100K,500K,1M} to ensure there are enough terminal nodes when sampling. We search the learning rates for all baselines among {3e-3,1e-3,3e-4,1e-4,3e-5,1e-5} and report the best performance. For our method, we fix the learning rate as 1e-4 and search our particular parameters. Value regularizer weight $\alpha \in \{0,0.2,0.5,0.8,1\}$, policy regularizer weight $\lambda \in \{0,0.01,1,2,5\}$ and temperature parameter $\tau \in \{1,0.1,0.01\}$.

**Evaluation.** We run each method on each environment with 20 different random seeds, and show the mean score and standard error with solid line and shaded area. The performance is evaluated by running 100 episodes after every 100K environmental steps. We use $\epsilon$-greedy exploration at evaluation with $\epsilon = 0.01$ to prevent the agent from being stuck at the same state.

Table 2: Hyper-parameters of DQN on Sokoban, MiniGrid and MinAtar environments.

| Hyperparameter | Value |
| --- | --- |
| Minibatch size | 32 |
| Replay memory size | 100,000 |
| Target network update frequency | 1,000 |
| Replay ratio | 0.25 |
| Discount factor | 0.99 |
| Optimizer | Adam |
| Initial exploration | 1 |
| Final exploration | 0.01 |
| Exploration decay steps | 1M |
| Total steps in environment | 2M |

## B  ENVIRONMENT DETAILS

### B.1  CLIFFWORLD

Cliffworld is a simple navigation task introduced by Sutton & Barto (2018) as shown in Figure 5. There are 48 states in total which is presented as two-dimensional coordinate axes $x$ and $y$. The size of action space is 4, with left, right, up and down. The agent needs to reach the goal state G starting from the start state S. The reward of reaching the goal is +1, dropped into the cliff gives -1, otherwise is 0. We set the discount factor as 0.99 and the max episode steps as 100. The arrow on the figure shows the optimal path.

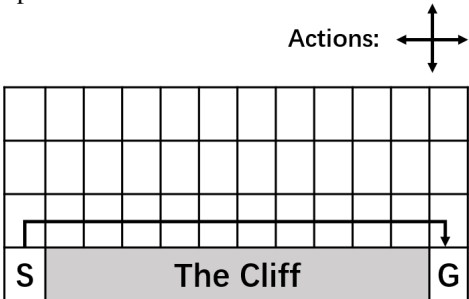

Figure 5: The illustration of Cliffworld environment. Each grid denotes a state, the arrow shows the optimal path from start state S to goal state G.

### B.2  SOKOBAN

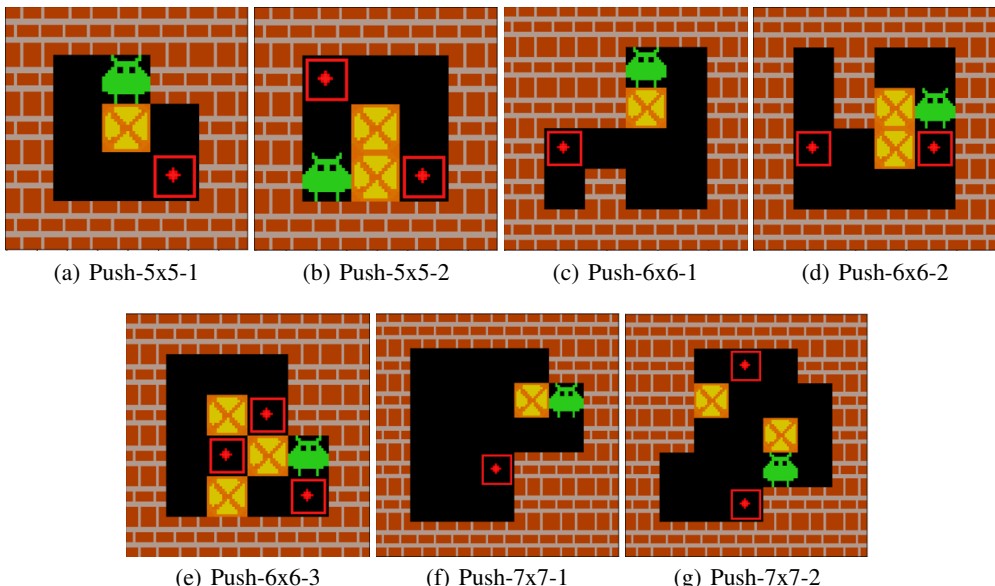

(a) Push-5x5-1          (b) Push-5x5-2          (c) Push-6x6-1          (d) Push-6x6-2

(e) Push-6x6-3          (f) Push-7x7-1          (g) Push-7x7-2

Figure 6: Visualization of Sokoban environments.

Sokoban (Junghanns & Schaeffer, 1997; Schrader, 2018) is a Japanese puzzle video game. The game is played on a board of squares, where each square is a floor or a wall. Some floor squares contain boxes, and some floor squares are marked as storage/target locations. The player needs to push all boxes on the target locations. The number of boxes equals the number of target locations. The puzzle is solved when all boxes are placed at target locations. The boxes cannot be pulled, it means a wrong action may lead to irreversible mistakes, which makes these puzzles so challenging especially for reinforcement learning algorithms, which mostly lack the ability to think ahead.

The room is randomly generated at each episode. Each time the environment is reset, the walls, the target locations, and the locations of the player and boxes will change. Therefore, it allows to train networks without overfitting on a set of predefined rooms. We choose seven tasks with increasing difficulties from Push-5×5-1 to Push-7×7-2. The number in the task name denotes the size of the map and the number of boxes. For example, 5x5-1 means a map with size 5×5 and 1 box to push. Figure 6 gives the visualization of the seven Sokoban tasks we choose.

**State Space.** We use an image-based state in our experiments. The state has three channels which denote RGB, the size of each channel is the same as the room size. Before inputting a state to the neural network, we divide it by 255 to normalize each pixel into $[0, 1]$.

**Action Space.** The action space is 8 in total, it consists of moving and pushing both in the 4 cardinal directions, left, right, up and down.

**Reward Function.** Pushing a box on the target location gets reward +1. Finishing the game gets reward +10. Each step will have a penalty reward of -0.1. And the max episode steps is 120. So in principle, the maximum reward we can achieve is near to 10 plus the number of boxes. And the minimum rewards is -12.

## B.3 MINIGRID

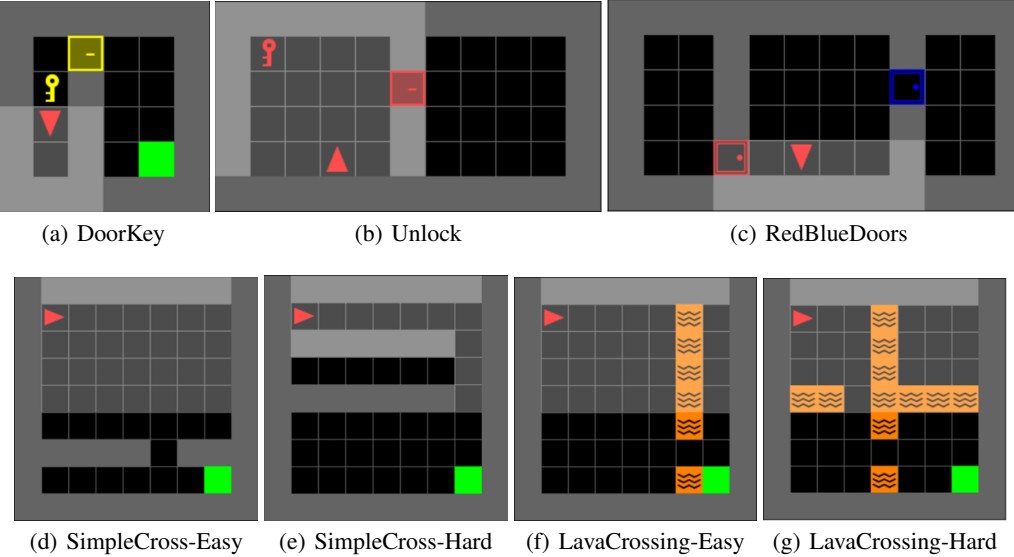

(a) DoorKey  (b) Unlock  (c) RedBlueDoors

(d) SimpleCross-Easy  (e) SimpleCross-Hard  (f) LavaCrossing-Easy  (g) LavaCrossing-Hard

Figure 7: Visualization of MiniGrid environments.

MiniGrid (Chevalier-Boisvert et al., 2018) is a gridworld Gymnasium (Brockman et al., 2016) environment, which is designed to be particularly simple, lightweight and fast. It implements many tasks in the gridworld environment and most of the games are designed with sparse rewards. We choose seven different tasks as shown in Figure 7.

The map for each task is randomly generated at each episode to avoid overfitting to a fixed map. The state is an array with the same size of the map. The red triangle denotes the player, and other objects are denoted with different symbols. The action space is different from tasks. For navigation tasks like SimpleCrossing and LavaCrossing, actions only include turn left, turn right and move forward. For other tasks like DoorKey and Unlock, actions also include pickup a key and open a door. Let MaxSteps be the max episode steps, MapWidth and MapHeight be the width and height of the map. We introduce each task as follows.

**DoorKey.** This task is to first pickup the key, then open the door, and finally reach the goal state (green block). MaxSteps is defined as $10 \times$ MapWidth $\times$ MapHeight. Reaching the goal state will get reward +MaxSteps/100, otherwise there is a penalty reward -0.01 for each step.

**Unlock.** This task is to first pickup the key and then open the door. MaxSteps is defined as $8 \times$ MapHeight$^2$. Opening the door will get reward +MaxSteps/100, otherwise there is a penalty reward -0.01 for each step.

**RedBlueDoors.** This task is to first open the red door and then open the blue door. MaxSteps is defined as $20 \times$ MapHeight$^2$. The agent will get reward +MaxSteps/100 after the red door and the blue door are opened sequentially, otherwise there is a penalty reward -0.01 for each step.

**SimpleCross-Easy/Hard.** This task is to navigate through the room and reach the goal state (green block). Knocking into the wall will keep the agent unmoved. MaxSteps is defined as $4 \times$ MapWidth $\times$ MapHeight. Reaching the goal state will get reward +MaxSteps/100, otherwise there is a penalty reward -0.01 for each step.

**LavaCross-Easy/Hard.** This task is to reach the goal state (green block). Falling into the lava (orange block) will terminate the episode immediately. MaxSteps is defined as $4 \times$ MapWidth $\times$ MapHeight. Reaching the goal state will get reward +MaxSteps/100, falling into the lava will get reward -MaxSteps/100, otherwise there is a penalty reward -0.01 for each step.

## B.4 MINATAR

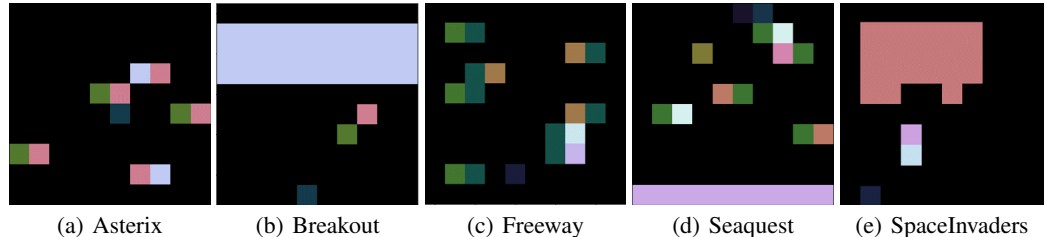

|  (a) Asterix  |  (b) Breakout  |  (c) Freeway  |  (d) Seaquest  |  (e) SpaceInvaders  |

Figure 8: Visualization of MinAtar environments.

MinAtar (Young & Tian, 2019) is image-based miniaturized version of Atari environments (Bellemare et al., 2013), which maintains the mechanics of the original games as much as possible and is much faster than original version. MinAtar implements five Atari games in total, we show the visualization of each game in Figure 8.

**State Space.** Each game provides the agent with a $10 \times 10 \times n$ binary state representation. The $n$ channels correspond to game specific objects, such as ball, paddle and brick in the game Breakout. The objects in each game are randomly generated at different time steps. The difficulty will change as the game progresses, for examples, there will be more objects and the objects will move faster. So these environments needs the policy to generalize across different configurations.

**Action Space.** The action space consists of moving in the 4 cardinal directions, firing, and no-op, and omits diagonal movement as well as actions with simultaneous firing and movement. This simplification increases the difficulty for decision making. In addition, MinAtar games add stochasticity by incorporating sticky-actions, that the environment repeats the last action with probability 0.1 instead of executing the agent's current action. This can avoid deterministic behaviour that simply repeats specific sequences of actions, rather than learning policies that generalize.

**Reward Function.** The rewards in most of the MinAtar environments are either 1 or 0. The only exception is Seaquest, where a bonus reward between 0 and 10 is given proportional to remaining oxygen when surfacing with six divers.

## C ADDITIONAL EXPERIMENTAL RESULTS

### C.1 OVERALL PERFORMANCE

We show the learning curves of each method on Sokoban, MiniGrid and MinAtar in Figure 9, 10 and 11. Each line is the average of running 20 different random seeds. The solid line shows the mean score and the shaded area shows the standard error. We can find our method CEER achieves

the best performance on most of environments, which indicates the conservative estimation helps the learning by a large margin. When the difficulty increases, the performance of CEER remains strong. Since Sokoban and MiniGrid are environments with sparse and delayed rewards, we conclude that our method is also stable on tasks with sparse and delayed rewards. This highlights that our methods is a general method that is suitable for all kinds of tasks.

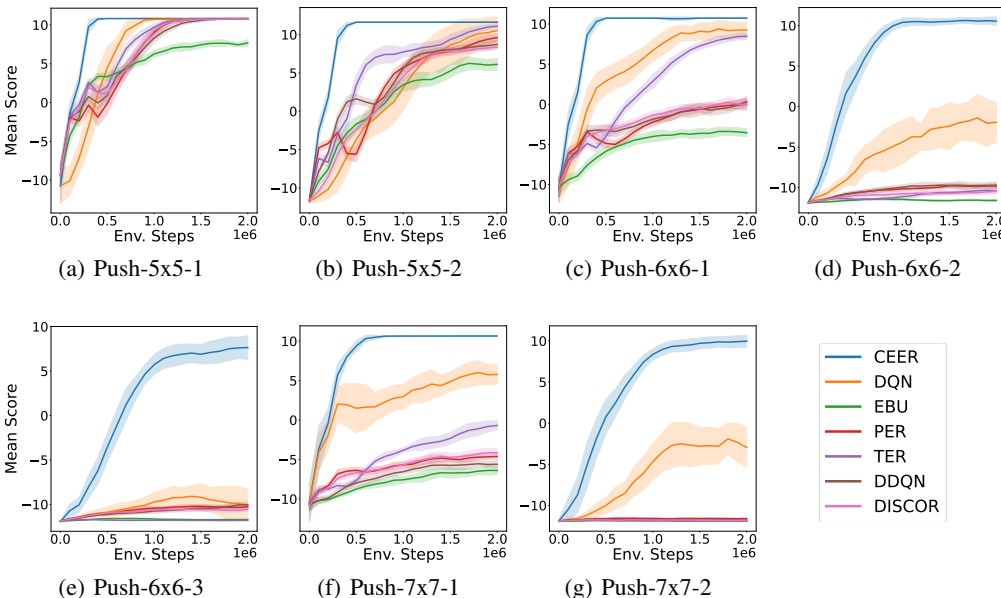

Figure 9: Learning curves on Sokoban environments. CEER outperforms other method by a large margin. When the difficulty increases, other methods failed while CEER still achieved high performance.

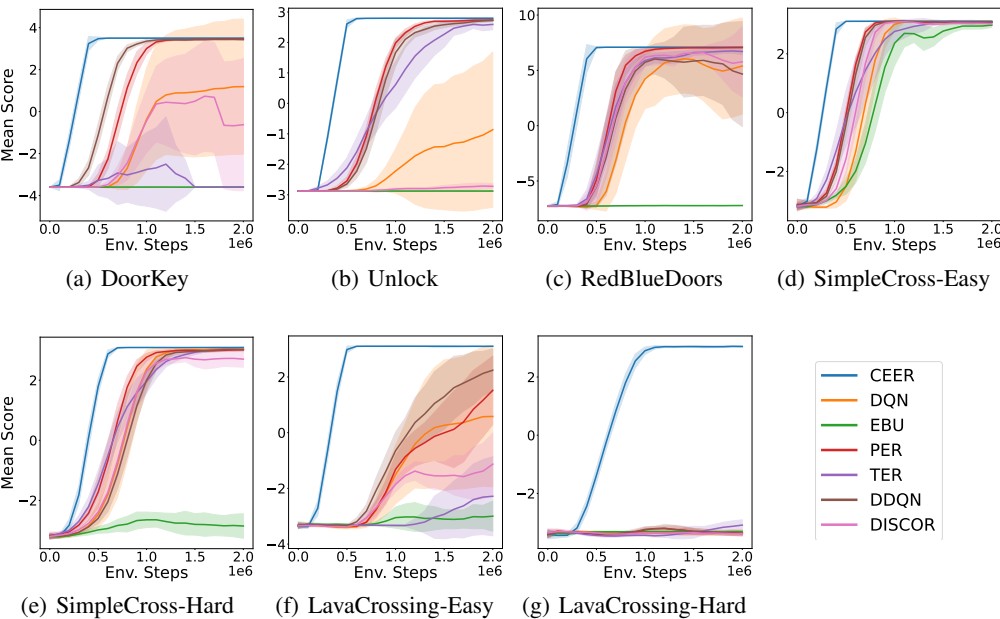

Figure 10: Learning curves on MiniGrid environments. CEER performs consistently well across all tasks, which shows our method is insensitive to sparse and delayed rewards environments.

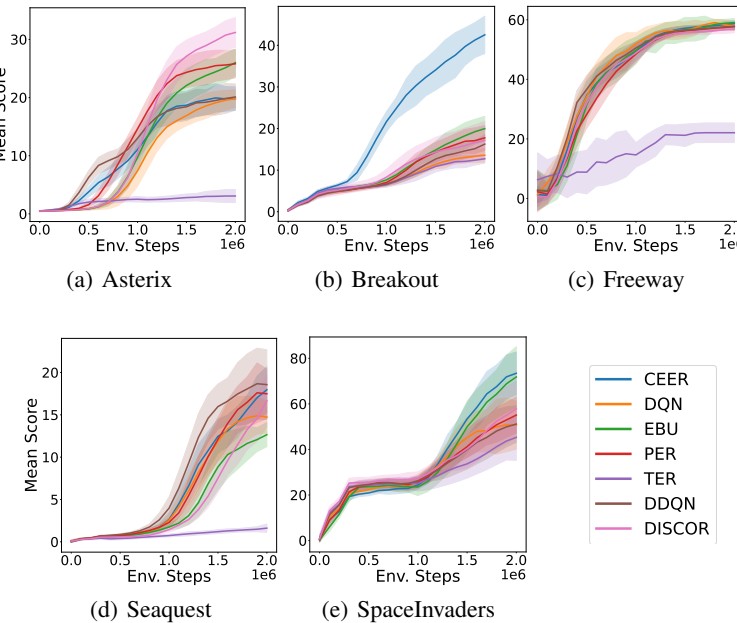

Figure 11: Learning curves on MinAtar environment. CEER performs best or achieves comparable performance in four out of five games. And on Breakout, the final score of CEER is twice more than other methods.

## C.2 Ablation Studies of Value and Policy Regularizers

We show the ablation studies of CEER on Sokoban, MiniGrid and MinAtar in Figure 12, 13 and 14. **Only_value** and **Only_policy** denote the methods that with only value or policy regularizer. CEER uses both regularizers and DQN uses no regularizer. **Conservative_policy** shows the result of the conservative policy derived from the replay memory. It uses the value regularizer with $\alpha = 1$ and maintains the policy regularizer unchanged.

We can find that our method, CEER, combines the benefits of both value and policy regularizers and achieves the highest final scores on almost all of the environments. **Only_policy** can achieve comparable performance as CEER on most of environments. It indicates the policy regularizer makes the most contribution for CEER. We can also find that the performance of **Only_value** and **Conservative_policy** is not consistent and even decrease on some environments. For examples, **Only_value** performs worse than DQN on environments like Push-6x6-1, SimpleCross-Hard and Asterix. **Conservative_policy** performs worse on environments like Asterix, Freeway and SpaceInvaders. Combining with the memory density of each environment as shown in Figure 3 and Table 3, we can find **Only_value** and **Conservative_policy** perform worse when the memory density decreases, especially for **Only_value**.

This is because only using value regularizer is not a direct regularization on learned policy, which may not maintain the correct order of actions. This emphasizes the importance of policy regularizer. And only learning from the conservative estimation is also not enough, especially on complex video games. The reason may because though the conservative estimation can help avoid dangerous actions that are never tried, it will also restrict exploring better actions. This highlights the necessary of learning from $Q$ network, which may take aggressive actions. We can consider the estimation of $Q$ network and the conservative estimation $\hat{Q}$ from RM-MDP as one kind of trade-off between exploration and exploitation. How to balance it is a promising topic which we leave for future work.

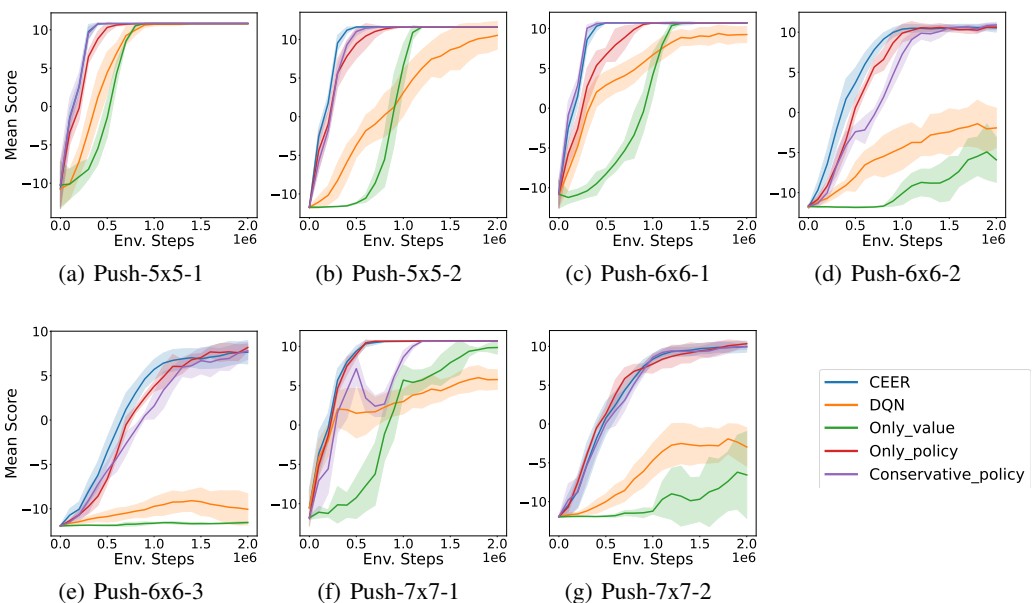

Figure 12: Ablation study on Sokoban environments. **Only_policy** achieves comparable performance as CEER. This indicates the policy regularizer makes the most contribution for CEER.

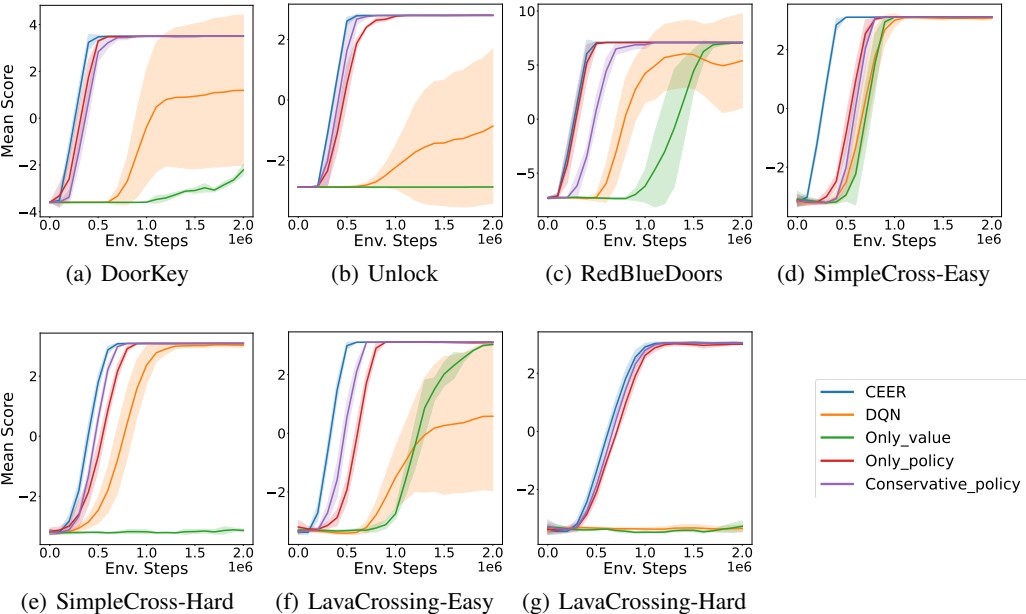

Figure 13: Ablation study on MiniGrid environments. **Only_value** fails on most of the tasks, which shows value regularizer is not a direct regularization on learned policy and cannot maintain the correct order of actions in some cases.

## C.3 MEMORY DENSITY AND ARPI FOR EACH ENVIRONMENT

We additionally show the statistics of memory density and ARPI for each environment in Table 3. Here, the memory density is the measurement of the RM-MDP's quality, and ARPI measures the relative performance improvement during learning. The definition can be found in Section 5.4. In Table 3, the memory density is the average during the whole learning, and the ARPI is the comparison between CEER and DQN.

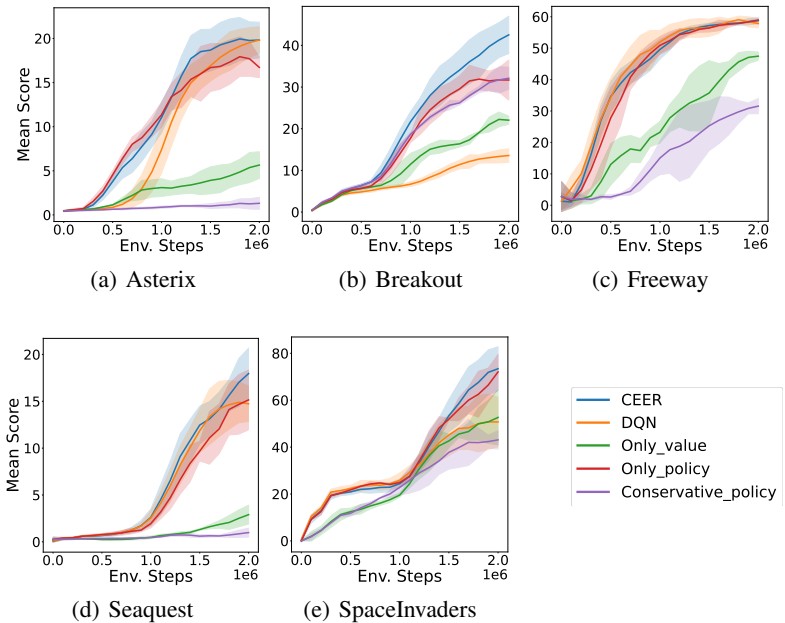

Figure 14: Ablation study on MinAtar environments. **Conservative_policy** performs poor on most of the environments. It shows though the conservative estimation can help avoid dangerous actions, it may also restrict exploring good actions.

On Seaquest environment, though the memory density is 0, we still find some improvement compared with DQN. This highlights that even the memory density is low, the conservative estimation can still help the learning. Freeway is the only exception that the performance decreases. The reason is that the speed of objects are not included into the state representation. The graph will merge different states into one vertex and provides wrong estimation, which is harmful for the learning. In addition, since the strong correlation between memory density and ARPI, memory density can be a measurement for the difficulty of environment, which is an interesting topic. For example, we can conclude that Push-6×6-3 is harder than Push-7×7-1 and Push-7×7-2.

Table 3: The memory density and ARPI on all environments during learning. The memory density is the average during learning, and ARPI shows the comparison between CEER and DQN.

| | Environment | Memory Density | ARPI (CCER\|DQN) |
|---|---|---|---|
| | Push-5×5-1 | 0.93 | 0.56 |
| | Push-5×5-2 | 0.84 | 0.76 |
| | Push-6×6-1 | 0.88 | 0.65 |
| Sokoban | Push-6×6-2 | 0.63 | 0.70 |
| | Push-6×6-3 | 0.53 | 0.66 |
| | Push-7×7-1 | 0.83 | 0.68 |
| | Push-7×7-2 | 0.56 | 0.64 |
| | DoorKey | 0.70 | 0.77 |
| | Unlock | 0.67 | 0.77 |
| | RedBlueDoors | 0.88 | 0.67 |
| MiniGrid | SimpleCrossing-Easy | 0.89 | 0.58 |
| | SimpleCrossing-Hard | 0.65 | 0.45 |
| | LavaCrossing-Easy | 0.86 | 0.75 |
| | LavaCrossing-Hard | 0.64 | 0.68 |
| | Asterix | 0.03 | 0.18 |
| | Breakout | 0.34 | 0.38 |
| MinAtar | Freeway | 0.02 | -0.08 |
| | Seaquest | 0.00 | 0.04 |
| | SpaceInvaders | 0.01 | 0.10 |

## C.4 Comparison With Model-based RL Methods

To make our work more solid, we compare our method with model-based methods. We summarize three kinds of algorithms, Dyna-style algorithms, Model-based Value Expansion (MVE-style) algorithms and Monte Carlo Tree Search (MCTS-style) algorithms. We combine the three paradigms with DQN algorithms, with perfect and learned model. Furthermore, we compare with DreamerV2 (Hafner et al., 2021), a state-of-the-art algorithms that learns powerful world models (Ha & Schmidhuber, 2018). Our results show that CEER is better or competitive with model-based methods with learned model and use much less wall-clock training time.

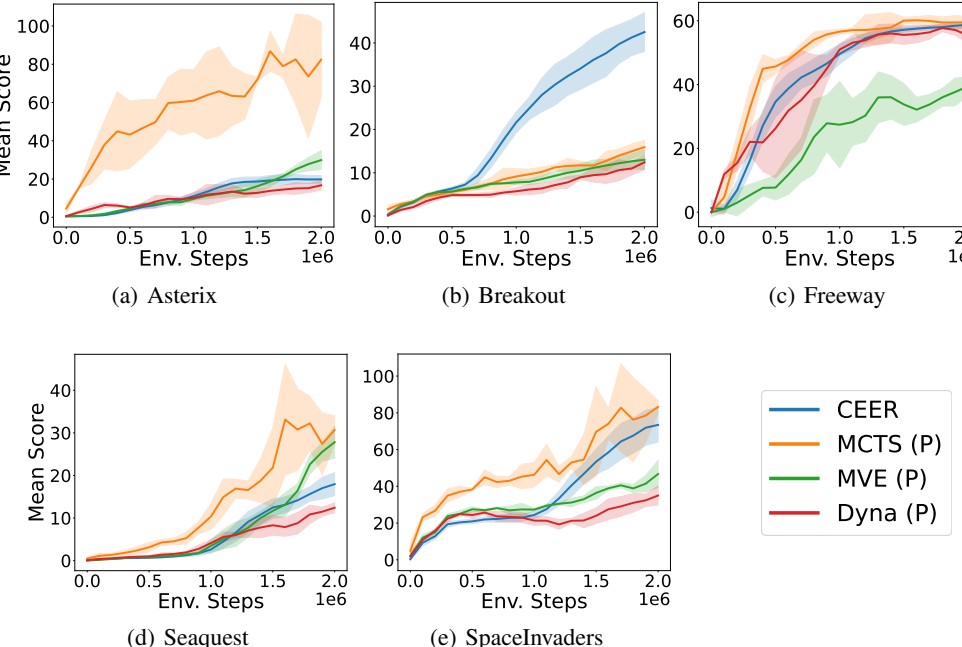

|   |   |   |
|---|---|---|
| (a) Asterix | (b) Breakout | (c) Freeway |
| (d) Seaquest | (e) SpaceInvaders | |

Figure 15: Comparison with model-based methods using perfect models. MCTS performs the best in four out of five games. This indicates MCTS is the most efficient model-based method even with small planning budgets.

**Dyna.** Dyna-style algorithms (Sutton, 1991; Sutton et al., 2012) achieve high sample efficiency by using the model to generate more transitions for training. Dyna first selects some start states from recent experience, then use a model to generate one-step transitions from those start states. It update the agent with both imaginary transitions and real transitions. In our implementation, besides an experience buffer saving experience from the environment, we additionally maintain a simulated experience buffer to save experience from the model. With each interaction step with environment, 5 start states are uniformly selected from 10,000 most recent states from experience buffer, then are used to simulate for one-step transitions using the model. When updates $Q$ networks, experiences are randomly sampled from both buffers.

**Model-based Value Expansion.** Model-free algorithms can improve value estimates by using $n$-step return and $\lambda$-return (Sutton & Barto, 2018). In MBRL, the model can be used for rollouts to improve the targets for temporal difference (TD) learning. This is known as model-based value expansion (MVE) (Feinberg et al., 2018; Buckman et al., 2018; Xiao et al., 2019). In particular, it computes $H$-step value estimates by simulating on-policy rollouts using the model. For real state $s_0$, a simulated trajectory with length $H$ is generated in the model following current policy, $(s_0, \tilde{a}_0, \tilde{r}_0, \tilde{s}_1, \cdots, \tilde{s}_{H-1}, \tilde{a}_{H-1}, , \tilde{r}_{H-1}, \tilde{s}_H)$. The state-value estimate sums up the component predicted by the model $\sum_{t=0}^{H-1} \gamma^t \tilde{r}_t$ and the tail estimated by $\hat{V}$,

$$\hat{V}_H(s_0) = \sum_{t=0}^{H-1} \gamma^t \tilde{r}_t + \gamma^H \hat{V}(\tilde{s}_H). \tag{11}$$

With $H$-step rollout, we have value estimates $(\hat{V}_0(s_0), \hat{V}_1(s_0), \cdots, \hat{V}_H(s_0))$. Then the target value is $Y_t = \frac{1}{H+1} \sum_{i=0}^{H} \hat{V}_i(s_0)$. Note that when $H = 0$, there is no rollout hence it recovers the model-free TD learning. The main advantages compared to model-free algorithms are the flexible on-policy estimates with the model. They do not require the importance weights of off-policy trajectories in model-free settings (Feinberg et al., 2018). We set $H$ as 5 in our implementation.

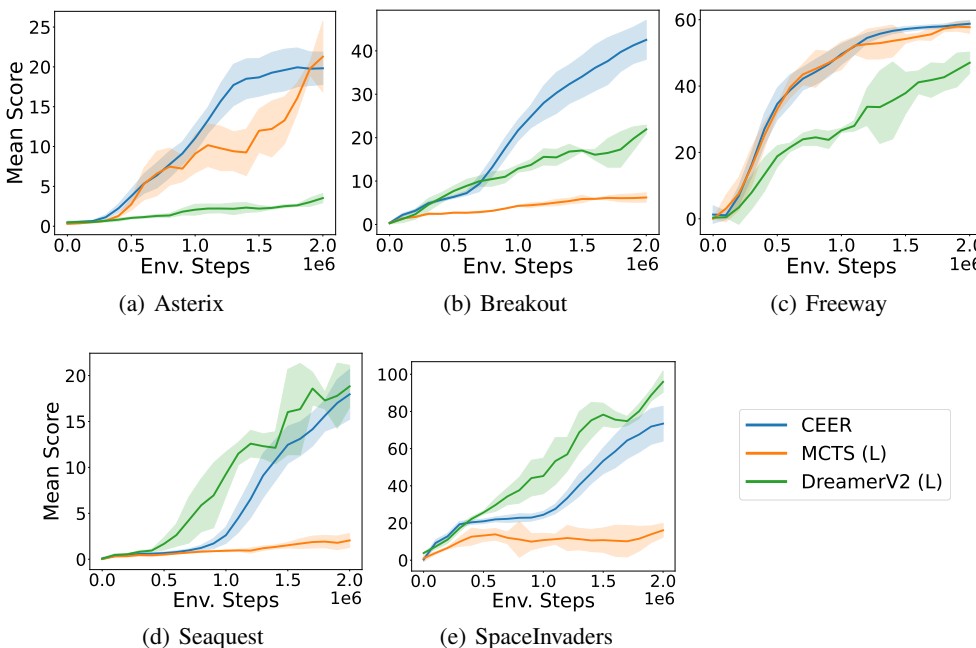

Figure 16: Comparison with model-based methods using learned models. CEER performs the best on Breakout and is also competitive with MVE and DreamerV2 on other games.

**Monte Carlo Tree Search.** The AlphaGo and AlphaZero algorithms alternately optimize a reactive (neural network) and a non-reactive (tree search) policy (Silver et al., 2016; 2017; 2018; Sun et al., 2018). Monte Carlo Tree Search (MCTS) (Kocsis & Szepesvári, 2006; Browne et al., 2012) incrementally builds a search tree to explore possible future states and actions. Each node in the search tree contains edges $(s, a)$ for all legal actions. Each edge stores statistics $\{N(s, a), W(s, a), \bar{Q}(s, a), P(s, a)\}$, where $N(s, a)$ is the visit count, $W(s, a)$ is the total action value, $\bar{Q}(s, a)$ is the mean action value and $P(s, a)$ is the prior probability of selecting that edge.

Each simulation consists of four phases: selection, expansion, evaluation, and backup. In the selection phase, actions are selected according to the statistics in the search tree to traverse from the root node to a leaf node. Two terms make up the selection formula,

$$a = \arg\max_a(\bar{Q}(s, a) + U(s, a)), \tag{12}$$

where $U(s, a) = c_{\text{puct}} P(s, a) \frac{\sum_b N(s, b)}{1 + N(s, a)}$ is an exploration term, $c_{\text{puct}} = 1.25$ is a parameter (Schrittwieser et al., 2020). Since there is no policy network in DQN, we transform $P(s, a)$ from the $Q$ network via softmax function $P(s, \cdot) = \mathbf{softmax}_\tau(Q(s, \cdot))$ with temperature parameter $\tau = 1$. In the expansion phase, the leaf node is expanded with edges. Each edge $(s, a)$ is initialized with statistics $\{N(s, a) = 0, W(s, a) = 0, \bar{Q}(s, a) = 0, P(s, a)\}$. In the evaluation phase, the new nodes are evaluated by network $Q(s, a)$. In the backup phase, statistics are updated backward. $N(s, a) = N(s, a) + 1, W(s, a) = W(s, a) + Q(s, a)$ and $\bar{Q}(s, a) = \frac{W(s, a)}{N(s, a)}$.

At each step, we only do five simulations to keep the training time within an acceptable range. In this case, comparing to selecting actions by visit counts after tree search, using action values is a more stable choice (Grill et al., 2020). Thus, after simulations by MCTS, we select actions based on the mean action values $\bar{Q}(s, \cdot)$ at the root node:

$$\pi(a|s) = \frac{\bar{Q}(s, a)^{1/\tau}}{\sum_b \bar{Q}(s, b)^{1/\tau}}. \tag{13}$$

Our implementation is a simplified version of AlphaZero and MuZero (Silver et al., 2017; Schrittwieser et al., 2020) that we don't learn a policy network. This is similar to Search with Amortized Value Estimates (SAVE) algorithm (Hamrick et al., 2020).

We first use perfect model (denote as P) for the three methods. Perfect model can avoid the difficulty of model learning and give us a clear insight which method is the best. Figure 15 shows the results comparing with CEER. We can find Dyna shows no advantage over other methods. This is because planning to generate one-step transitions provides limited data diversity compared to an existing experience buffer, which offers little benefit even with a perfect model (Holland et al., 2018; Hasselt et al., 2019). This also gives us a similar results to Van Hasselt et al. (2019) that replay-based algorithms should be competitive to or better than model-based algorithms if the model is used only to generate fictional transitions. MVE performs better than CEER on Asterix and Seaquest but performs worse on other games. We hypothesize the choice of $H$ has a trade-off between variance and bias, which is similar to $n$-step return and depends on the specific problem. MCTS performs the best in four out of five games. This indicates when we have a perfect model, MCTS is the most efficient method even with a small planning budgets.

Since perfect model is not an practical setting in real life. Next, we compare with methods that learn a model (denote as L). We consider MCTS with learned model, and the state-of-the-art baseline DreamerV2 (Hafner et al., 2021). For the learned model in MCTS (L), we use the same architecture as $Q$ network. The input is state $s$ and action $a$, the output is the next state $s'$ and reward $r$. DreamerV2 is a method that focuses more on how to learn a high-quality model. We maintain the architecture as in original paper. There are too many parameters in DreamerV2, which is more complex and harder to tune than CEER. Figure 16 shows that CEER performs the best on Breakout, and is also competitive with DreamerV2 and MCTS (L) on other games.

## C.5 WALL-CLOCK TIME COMPARISON

Only reporting sample efficiency cannot tell us how long we need to train an agent for each method. We compare the wall-clock time during training in Table 4. We use *Frames Per Second* (FPS) to measure the training speed. FPS counts the number of frames that the agent interacts with the environment per second. We test the speed on device with 1 GPU (NVIDIA TITAN RTX) and 10 CPUs (Intel(R) Xeon(R) CPU E5-2650 v4 @ 2.20GHz). We do 3 runs for DreamerV2 and 5 runs for other methods. We can find our method CEER (FPS: 308) is only slightly slower than DQN (FPS: 402), similar to PER (FPS: 318), and much faster than model-based methods like DreamerV2 (FPS: 19). Since we use a simple network architecture with a single convolutional layer followed by a fully connected layer, the computation burden for the memory part is relatively high comparing with the network training. If we use a large network, the computation burden for the memory part will be lower comparing with the network training.

Table 4: Wall-clock time comparison between different model-free and model-based methods. We use *Frames Per Second* (FPS) to measure the speed of interaction with environments during training. For model-based methods, **P** denotes the method using a perfect model, **L** denotes the method that learns a model.

| Method | | **FPS** (mean $\pm$ std) |
|---|---|---|
| | DQN | $402.85 \pm 6.78$ |
| | DDQN | $388.73 \pm 5.97$ |
| | PER | $318.17 \pm 1.43$ |
| Model-Free | EBU | $193.60 \pm 4.28$ |
| | DisCor | $328.43 \pm 5.86$ |
| | TER | $220.84 \pm 3.50$ |
| | CEER | $308.72 \pm 5.68$ |
| | Dyna (P) | $117.67 \pm 0.81$ |
| | MVE (P) | $79.09 \pm 2.22$ |
| Model-Based | MCTS (P) | $104.78 \pm 0.51$ |
| | MCTS (L) | $50.21 \pm 0.90$ |
| | DreamerV2 (L) | $19.36 \pm 0.88$ |

### C.6 Additional Comparison with Rainbow on Atari Pong Environment

Besides MinAtar environments, we additionally compare our method with Rainbow (Hessel et al., 2018) algorithm on full suite of Atari game Pong (Bellemare et al., 2013). The network architecture contains three convolutional layers and a fully connected hidden layer, which is the same as used in DQN (Mnih et al., 2015). Rainbow splits the Q values into state-value and the advantages for each action. We show the result on Pong in Figure 17. We can find CEER learns faster than Rainbow, which indicates CEER is competitive on the full suite of Atari environments.

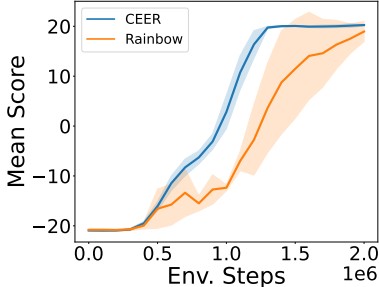

Figure 17: Learning curves on Atari Pong environment. CEER learns faster than Rainbow, which indicates CEER is also competitive on the full suite of Atari environments.

### C.7 Comparison between Sampling Update with Sweeping Update

Since each interaction step will add a new transition to the memory during the training, RM-MDP is non-stationary. A natural question is if the sampling-based update and episode-based update (Sampling Update for short) introduced in Section 4.2 is enough to obtain accurate estimation? Another time-consuming but completely accurate update is to sweep all state-action pairs in the memory at each interaction step (Sweeping Update). We compare the two ways on three environments as shown in Figure 18. We can find though RM-MDP is non-stationary, CEER with Sampling Update perform almost the same as Sweeping Update, which means Sampling Update is enough to obtain accurate estimation in this non-stationary MDP.

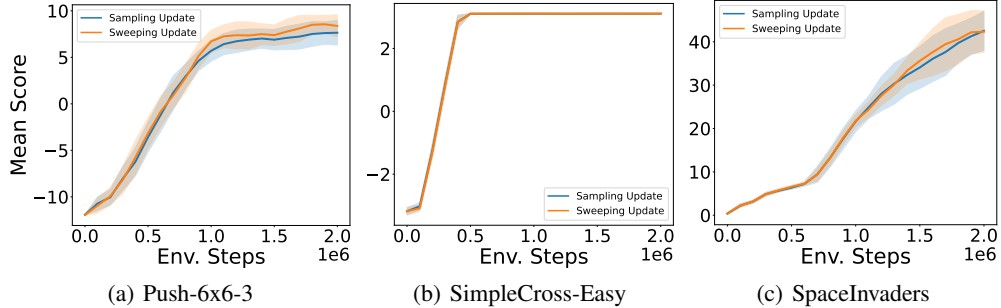

| (a) Push-6x6-3 | (b) SimpleCross-Easy | (c) SpaceInvaders |

Figure 18: Comparison between Sampling Update with Sweeping Update. CEER with Sampling Update performs almost the same as Sweeping Update, which means Sampling Update can obtain accurate estimation even in this non-stationary MDP.

### C.8 Parameter Search for Value and Policy Regularizers

We show the parameter search for value ($\alpha$) and policy ($\lambda$) regularizers in Figure 19 and 20. When we report one parameter, we fix the other parameter as the optimal one. The vertical axis shows the final score. We can find when the environments are easy,such as Push-5x5-1 and SimpleCrossing-Easy, the two regularizers do not influence the final performance. When the environments are harder, such as Push-6x6-3 and Breakout, the two regularizers play important role in performance improvement, which emphasizes the efficiency of our method.

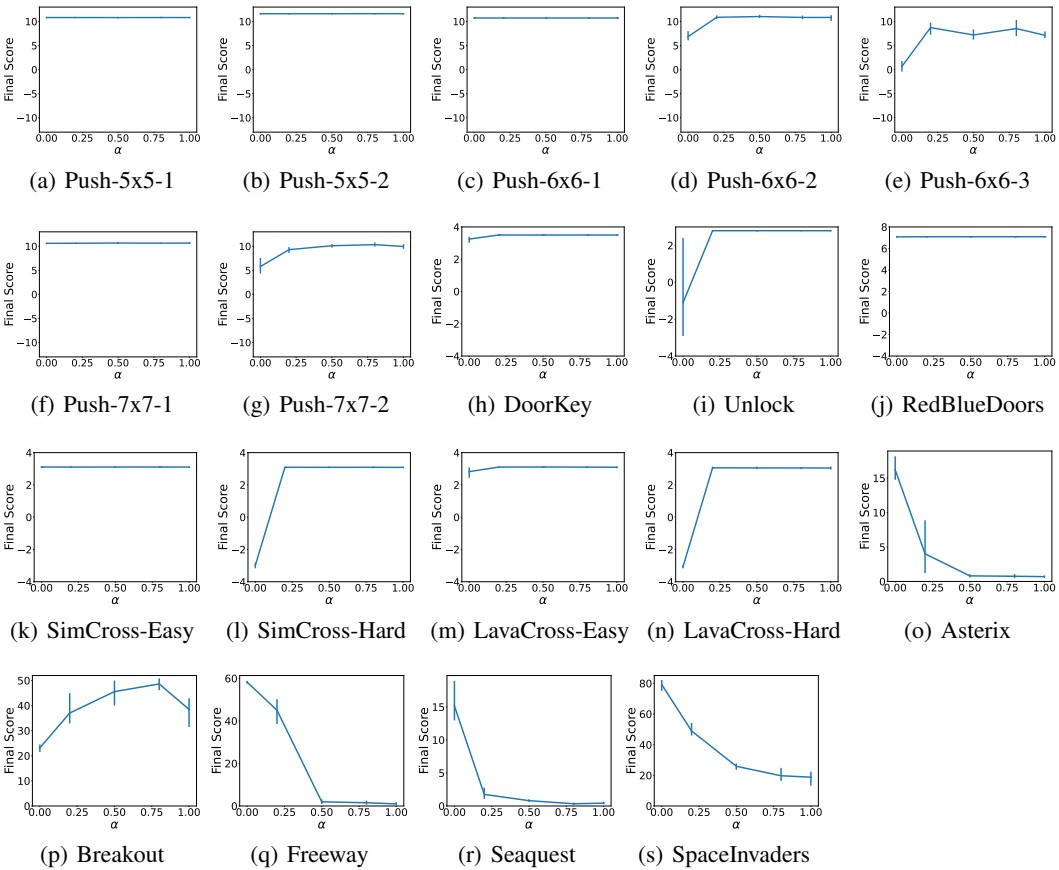

Figure 19: Parameter search for value regularizer.

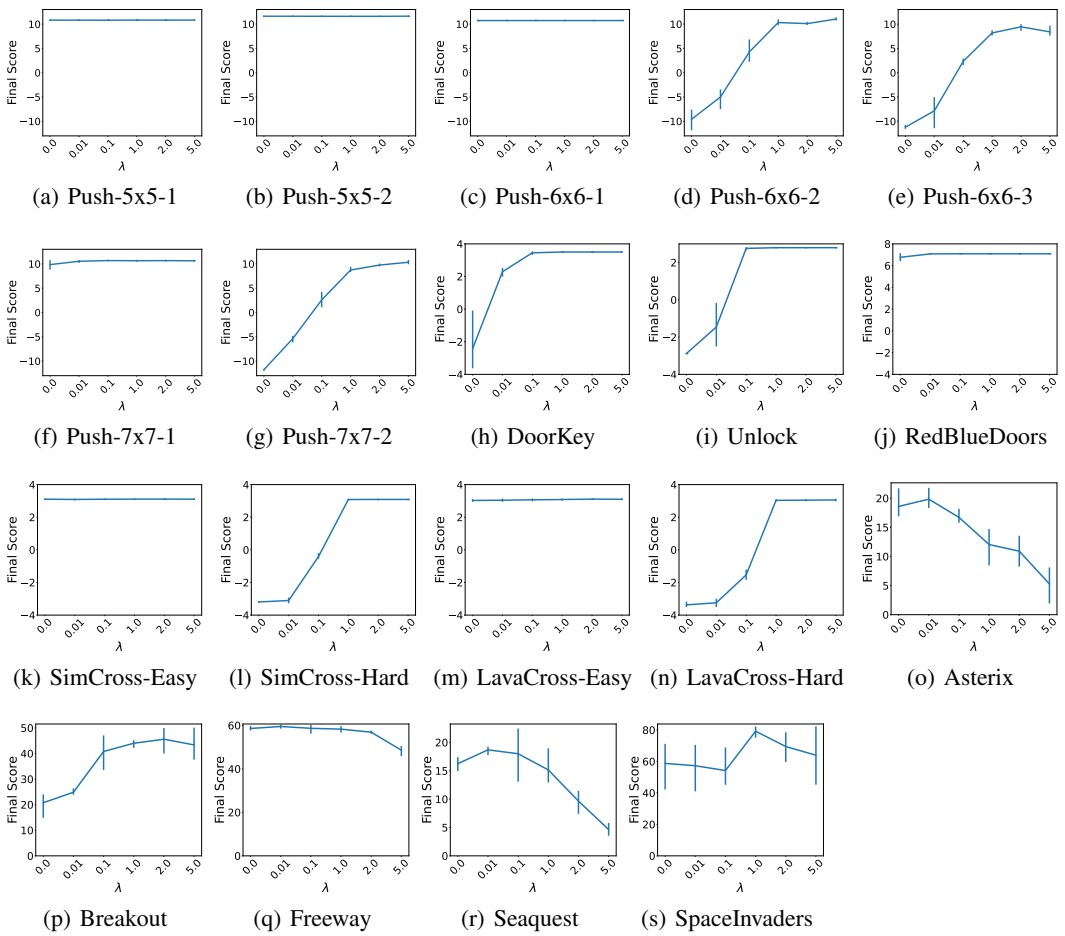

Figure 20: Parameter search for policy regularizer.

