# OpenReview forum: "Replay Memory as An Empirical MDP: Combining Conservative Estimation with Experience Replay"
_ICLR.cc/2023/Conference — ICLR 2023 poster_

### Official Review · Reviewer_oYzW · 2022-10-19

**Confidence:** 4
**Clarity, Quality, Novelty And Reproducibility:** In Strength And Weaknesses
**Correctness:** 3
**Technical Novelty And Significance:** 3
**Empirical Novelty And Significance:** 3
**Recommendation:** 8

**Strength And Weaknesses:**

Summary: This paper considers constructing an empirical MDP from experience replay and estimates a Q-value function and policy from this empirical MDP. Then this paper applies these estimators to help train original Q-network. The benefit of the empirical MDP on replay buffer is the conservative estimation, which helps accelerate the learning process of orinigal Q-network.

Overall, I reckon the idea of this paper is quite novel and interesting, and this paper provides convincing empirical studies of their algorithms. However, there existing some writing issues and some further questions to me in the following. At the first stage, I would give a conservative evaluation of this paper with the score of 6. I will consider change my score with other reviewers' opinions and authors' responses.

Questions and Comments:
1. Eqn (1), the LHS is Q(s,a) but the RHS is the expectation over (s,a,s'). I believe it's a typo?
2. Eqn (3), what does the 't' mean for (s_t, a_t, s_t+1)?
3. In Data Structure of Section 4.1, the main purpose is to estimate the empirical transition probabilities by frequency estimation. My question is what is the role of the Graph? Without the Graph, it seems the estimation can also be derived.
4. I think the authors should be more specific on the expression of the sampling method in Section 4.2. Though I can understand what the authors are trying to say, others may not.
5. The authors should offer references on the overestimation problem on DQN in paragraph "value regularize".
6. Could the authors tell me more about how conservative policy $\hat{\pi}$ lower bounds the performance of policy $\pi$.
7. In algorithm 1, do updates in Line 9 and Line 10 share the sample minibatch? Why？
8. Actually, I'm curious about the choice of $\alpha$ in value regularize. It would be great if the author compare the performances of their algorithm with different choice of alphas.
9. To my knowledge, the model-free methods mean the memory space is taken up to $\mathcal{O}(SA)$. However, by constructing empirical MDP with estimation of transition probability, the memory space is taken up to $\mathcal{O}(S^2A)$, which should be model-based. I think the authors should be more precise on description of their methods in introduction and abstract.
10. In Section 5.4, the authors investigate the relationship between the performance improvement and conservative estimation. But I'm still confused with the intuition why density is correlated with the performance improvement. Could the authors provide a more high-level explanation in rebuttal?

==========================

After reading the authors responses and other reviewers' comments, I find the current version is qualified to be published. At the current stage, I prefer to raise my score to 8. If the other reviewers have more concerns, I'm willing to discuss with them and adjust my evaluation.

**Summary Of The Paper:**

In Strength And Weaknesses

**Summary Of The Review:**

In Strength And Weaknesses

---

> ### Author Response · Authors · 2022-11-12
> **Response to Reviewer oYzW**
>
> Thank you for the review! We answer the questions below. Please let us know if you have any further questions or suggestions. We have also fixed the typos that you pointed out in the new version.
>
> > - In Data Structure of Section 4.1, the main purpose is to estimate the empirical transition probabilities by frequency estimation. My question is what is the role of the Graph? Without the Graph, it seems the estimation can also be derived.
>
> The purpose of the graph is to merge states repeated in different trajectories into one node and record the visit count $N(s,a,s^\prime)$, which is used to estimate $\hat{p}(s^\prime|s,a)$. The benefit is that with the graph, we can record the statistics such as $r, \hat{Q},\hat{P}$ and query them in $\mathcal{O}(1)$ time, which is efficient.
>
> > - I think the authors should be more specific on the expression of the sampling method in Section 4.2. Though I can understand what the authors are trying to say, others may not.
> > - The authors should offer references on the overestimation problem on DQN in paragraph "value regularize".
>
> Thank you for your careful reading. We fixed these problems in the new version.
>
> > - Could the authors tell me more about how conservative policy $\hat{\pi}$ lower bounds the performance of policy $\pi$.
>
> The policy $\hat{\pi}$ is estimated based on transitions in the memory. If a state/action is not in the memory, it means there is no node/edge for that state/action, and the transition probability is 0. This is conservative, we only use information from experienced transitions. $\hat{\pi}$ may not be the optimal policy for the whole environment, but it provides us with a policy that performs well on some part of the environment. And if we regularize policy $\pi$ with policy $\hat{\pi}$, $\pi$ should perform at least as well as $\hat{\pi}$. That is why we say $\hat{\pi}$ lower bounds the performance of $\pi$.
>
> > - In algorithm 1, do updates in Line 9 and Line 10 share the sample minibatch? Why？
>
> Yes, since DQN will sample a batch every four environmental steps, we use the same batch to update $\hat{Q}$ and the $Q$ network. So there is no extra computation for sampling. We added an experiment in Appendix C.7 Figure 18 to show that this Sampling Update can achieve almost the same performance as a Sweeping Update (update all transitions in the memory every four steps). It is enough to obtain accurate estimation and also saves computation.
>
> > - Actually, I'm curious about the choice of $\alpha$ in value regularize. It would be great if the author compare the performances of their algorithm with different choice of alphas.
>
> Thank you for the suggestion. We provide the parameter search for value regularizer $\alpha$ and policy regularizer $\lambda$ in Appendix C.8 Figure 19, 20.
>
> > - To my knowledge, the model-free methods mean the memory space is taken up to $\mathcal{O}(SA)$. However, by constructing empirical MDP with estimation of transition probability, the memory space is taken up to $\mathcal{O}(S^2A)$, which should be model-based. I think the authors should be more precise on description of their methods in introduction and abstract.
>
> From the theory perspective, the memory space will be $\mathcal{O}(S^2A)$. From the implementation, since all statistics are scalars except the representation of states $s$ (picture), the memory consumption is dominated by storing states.
> We implement the memory as a fixed length queue of states and statistics, that stores the most recent experiences. This is the same as for DQN memory.  So the memory consumption is the same in practice. Only a small subset of the state space is stored in memory at any time.
>
> > - In Section 5.4, the authors investigate the relationship between the performance improvement and conservative estimation. But I'm still confused with the intuition why density is correlated with the performance improvement. Could the authors provide a more high-level explanation in rebuttal?
>
> The memory density is a metric to measure the quality of RM-MDP. If the memory density is higher, RM-MDP will be a closer match to the full environment MDP, leading to a better estimation $\hat{Q}$. With this better estimation as regularizers to lower bound the $Q$ network, the performance tends to improve. If the memory density is low, there is no intersection between trajectories. The graph degenerates to separate connected components equivalent to the replay memory used in DQN, and the conservative estimation degenerates to $n$-step return. In this case, the regularizers help little. That is the reason why the memory density is correlated with the performance improvement.

---

> > ### Comment · Reviewer_oYzW · 2022-11-16
> > **Response**
> >
> > I would like to thank the authors for addressing my comments and suggestions. My significant concerns have been addressed satisfactorily and the revised version is much easier to read. I would tend to raise my score and confidence if the authors could address the following minor concerns.
> >
> > 1. Could the authors give me more explanation on the performance of the conservative estimator? For example, should I expect the conservative estimator performs as well as the Q value function at the final stage?
> >
> > 2. I'm also wondering what is the difference between memory density and coverage rate in offline RL. It seems a high memory density is similar to a high coverage rate in offline RL.
> >
> > 3. From the Fig. 4, I find that the performance of Only_value is much worse than Only_policy. Could the authors give me some explanation of this phenomenon?

---

> > > ### Author Response · Authors · 2022-11-17
> > > **Response to Reviewer oYzW's Further Concerns**
> > >
> > > We thank reviewer oYzW for your positive support. We provide explanations below.
> > >
> > > > 1. Could the authors give me more explanation on the performance of the conservative estimator? For example, should I expect the conservative estimator performs as well as the Q value function at the final stage?
> > >
> > > Thank you for this interesting question. One limitation of the conservative estimator is that it is only defined for states inside the replay memory. It does not try to generalize to states outside the memory, and it will return the best possible estimator that is limited to the memory. Inside the memory, the estimator can avoid bad/unsafe actions, but it can also miss optimal actions that lead to high rewards. When we combine it with $Q$-network, we alternately optimize two value estimates, $\hat{Q}$ and $Q$. This is similar to alternately optimizing two policies [1]. An $\epsilon$-greedy policy using $Q$ interacts with the environment and collects data into the memory. This helps learn a better $\hat{Q}$ based on the memory. And the $\hat{Q}$ regularizes $Q$ in a supervised manner to stabilize TD learning. In this sense, when the learning converges, both $\hat{Q}$ and $Q$ may become very similar for the in-memory states based on seeing the same data and thus perform similarly. The experiments on MiniGrid and Sokoban shown in Appendix C.2 Figure 12, 13 show some evidence.
> > >
> > > [1] Sun, Wen, et al. "Dual policy iteration." Advances in Neural Information Processing Systems 31 (2018).
> > >
> > > > 2. I'm also wondering what is the difference between memory density and coverage rate in offline RL. It seems a high memory density is similar to a high coverage rate in offline RL.
> > >
> > >
> > > The memory density only focuses on actions for in-memory states (see Equation 10). On the other hand, the coverage rate in offline RL considers how many transitions of the original MDP have been covered by the data set. To see their difference for example, if the replay memory contains a small number of states that have tried many actions, the memory density is high but the coverage rate is low.
> > >
> > > > 3. From the Fig. 4, I find that the performance of Only_value is much worse than Only_policy. Could the authors give me some explanation of this phenomenon?
> > >
> > > Our conjecture is that although the value regularizer could reduce the bias of the value estimation, the correct order of actions might not be maintained. For example, assume there are two actions $a_1, a_2$ at a state $s$ with oracle optimal values $q(s,a_1)=1$ and $q(s,a_2)=0.99$. $a_1$ is the optimal action. A $Q$-network is initialized with $Q(s,a_1)=0$ and $Q(s,a_2)=0$. After some updates, we may have $Q(s,a_1)=0.99$ and $Q(s,a_2)=1$ due to the function approximation error induced by the neural network with gradient-based optimization. In this case, though the mean square error is reduced, the correct order of actions is also changed. By comparison, the policy regularizer is a direct regularization of policy, which is not affected by such issue.

---

> > > ### Author Response · Authors · 2022-11-22
> > > **Thank you for your comments and feedback**
> > >
> > > We are glad that the reviewer's concerns have been addressed. We thank the reviewer oYzW for the time and constructive feedback.

---

### Official Review · Reviewer_nSvQ · 2022-10-22

**Confidence:** 5
**Correctness:** 3
**Technical Novelty And Significance:** 2
**Empirical Novelty And Significance:** 2
**Recommendation:** 8

**Clarity, Quality, Novelty And Reproducibility:**

I do not have concerns about clarity, quality, or reproducibility. The novelty is unclear since the paper is similar to many works that combine model-based and model-free algorithms.

**Details Of Ethics Concerns:**

No concern.

**Strength And Weaknesses:**

Strengths:

1. The motivation is clear. The algorithm is interesting.
2. The authors show improvement in sample efficiency in a toy environment and some more complex environments in various domains.

Weaknesses:

1. While the paper focuses on the experience repay, I find the idea is actually to combine model-based and model-free algorithms. Several related methods should be discussed and compared, such as World Models [1] [2], and works that combine model-based and model-free methods, such as [3]. That being said, it is unclear what the novelty is without a comparison with these previous works.
2. Some missing related works in experience replay [4] [5].
3. I find the idea is very similar to ERLAM [6]. It seems the only difference is the way the regularizer is added.
4. In Eq. (3), the transition probabilities are estimated by visiting counts. In the case that a state has never been seen or an action has never been tried by the agent, how should the probabilities be estimated?
5. Can the methods be used in continuous state space? In this case, how to construct the graph, and how the value iteration can be applied?
6. The paper only reports sample efficiency but not wall-clock time. It is unclear how many additional computational resources are needed for the graph construction and value iteration.
7. How the memory density in Eq. (10) could be applied in continuous state space.




[1] Ha, David, and Jürgen Schmidhuber. "World models." arXiv preprint arXiv:1803.10122 (2018).

[2] Hafner, Danijar, et al. "Dream to Control: Learning Behaviors by Latent Imagination." International Conference on Learning Representations. 2019.

[3] Sun, Wen, et al. "Dual policy iteration." Advances in Neural Information Processing Systems 31 (2018).

[4] Zha, Daochen, et al. "Experience replay optimization." Proceedings of the 28th International Joint Conference on Artificial Intelligence. 2019.

[5] Fujimoto, Scott, David Meger, and Doina Precup. "An equivalence between loss functions and non-uniform sampling in experience replay." Advances in neural information processing systems 33 (2020): 14219-14230.

[6] Zhu, Guangxiang, et al. "Episodic reinforcement learning with associative memory." (2020).


**Summary Of The Paper:**

This paper presents an idea of combining conservative estimation with experience replay to improve sample efficiency. The key idea is to construct a graph based on the transitions stored in the buffer, where the transition probabilities are estimated based on the visiting counts. Then they use value iteration to solve the graph (i.e., an estimated MDP) to obtain the optimal Q-value for the estimated MDP. The obtained Q-value is further derived to an optimal policy. Finally, the optimal Q-values and policy are used to regularize the policy and value networks, where the optimal Q-values are combined with the RL Q-values with the weighted sum, and the optimal policy network is used as the target of policy distillation with KL divergence loss. Experimental results on several environments show that the proposed method CEER outperforms the baselines.

**Summary Of The Review:**

The paper is well-motivated and presents an interesting idea. However, the paper is similar to many works that combine model-based and model-free algorithms. Their relationships have not been discussed. Also, the method may not be able to scale to large or continuous state space.

---

> ### Author Response · Authors · 2022-11-12
> **Response to Reviewer nSvQ, Part 1**
>
> Thank you for the review. We are glad to answer the questions below. Please let us know if this fully addresses your concerns, or if there are other issues that we should address. We have also fixed the typos in the new version.
>
> > - While the paper focuses on the experience repay, I find the idea is actually to combine model-based and model-free algorithms. Several related methods should be discussed and compared, such as World Models [1] [2], and works that combine model-based and model-free methods, such as [3]. That being said, it is unclear what the novelty is without a comparison with these previous works.
>
> Thank you for this constructive suggestion! We found two disadvantages of model-based approaches that made us focus more on how to make full use of existing transitions in the replay memory. First, while model-based methods with a perfect model can achieve better performance than model-free methods, learning a good model can be difficult. Second, the wall-clock training time increases dramatically with model learning and model planning.
>
> Comparing with model-based methods, the benefit is that the transitions in the replay memory are accurate, and this kind of non-parametric model is easy to obtain. We can achieve good performance if we use replay memory properly [1].
> To make our work more convincing, we add experiments comparing with model-based methods Dyna [2], MVE [3], MCTS [4][5][6], and DreamerV2 [7] in Appendix C.4 Figure 15, 16., and show a wall-clock time comparison in Appendix C.5 Table 4.
> Our method CEER is still better or competitive with model-based methods that learn a model. We also add discussions in Section 2 Related Work.
>
> [1] Van Hasselt, Hado P., Matteo Hessel, and John Aslanides. "When to use parametric models in reinforcement learning?." Advances in Neural Information Processing Systems 32 (2019).
>
> [2] Sutton, Richard S. "Dyna, an integrated architecture for learning, planning, and reacting." ACM Sigart Bulletin 2.4 (1991): 160-163.
>
> [3] Feinberg, Vladimir, et al. "Model-based value estimation for efficient model-free reinforcement learning." arXiv preprint arXiv:1803.00101 (2018).
>
> [4] Silver, David, et al. "Mastering the game of go without human knowledge." nature 550.7676 (2017): 354-359.
>
> [5] Hamrick, Jessica B., et al. "Combining Q-Learning and Search with Amortized Value Estimates." International Conference on Learning Representations. 2020.
>
> [6] Sun, Wen, et al. "Dual policy iteration." Advances in Neural Information Processing Systems 31 (2018).
>
> [7] Hafner, Danijar, et al. "Mastering Atari with Discrete World Models." International Conference on Learning Representations. 2021.
>
> > - I find the idea is very similar to ERLAM [6]. It seems the only difference is the way the regularizer is added.
>
> Besides the regularizers we designed, there are other differences and contributions in our work:
> ERLAM maintains a second graph memory to store good past experiences, and updates the values of the graph in a reversed breadth-first manner. This way is not suitable for stochastic environments, and requires techniques to handle cycles in the graph.
> In contrast, our method directly builds a graph-based replay memory, which constructs an empirical MDP over the collected transitions. We then solve this MDP through a modified dynamic programming scheme.
> Our method is therefore more general and has no constraints on the environment MDP. For example, it can be stochastic and can have cycles.
>
> Additionally, from our experiments, we find a strong correlation between memory density and performance improvement. This relationship gives us a clear understanding of the circumstances when our method will help most. This strong relationship definitely can guide future work to diagnose and design new methods.
>
> In the high-level view, our method focuses on conservative estimation from replay memory. Constructing a graph is a way to obtain such a conservative estimate, but it is not the only way. There are many offline RL methods [2] for solving the empirical MDP which result in a conservative estimate. Our method shows a path for learning from data in an online replay memory by using offline static dataset methods, which we think is a promising direction for further future work.
>
> [1] Zhu, Guangxiang, et al. "Episodic Reinforcement Learning with Associative Memory." International Conference on Learning Representations. 2019.
>
> [2] Levine, Sergey, et al. "Offline reinforcement learning: Tutorial, review, and perspectives on open problems." arXiv preprint arXiv:2005.01643 (2020).

---

> > ### Comment · Reviewer_nSvQ · 2022-11-21
> > **Thank you for the response, one question needs further clarification**
> >
> > Most of my concerns are resolved. However, I am still not clear how the methods could work on continuous (or high-dimensional) state space. The authors only discuss continuous "action" space but not state space. While discrete action space is common and it is fine to defer continuous action space to future work, most of the environments have continuous (or high-dimensional) state space, e.g., Atari games, or any pixel inputs. Thus, being able to deal with continuous state space is very important. I would like to ask the authors to clarify.

---

> > > ### Author Response · Authors · 2022-11-21
> > > **Response to Reviewer nSvQ's further question**
> > >
> > > > I am still not clear how the methods could work on continuous (or high-dimensional) state space.
> > >
> > >
> > > **In our experiments, MinAtar environments are pixel inputs with high-dimensional state space (10x10xn channels).** We use hashing function to map the state to a key and merge states with the same key. In this case, nearly no state can be merged and the memory density is low. Surprisingly as shown in Section 5.4 Figure 3, there are still obvious improvements. This indicates even if the memory density is low due to high or continuous state space, there is still an obvious improvement.
> > >
> > > For example, in Figure 3, the memory density for **SpaceInvaders (10x10x6)** is near 0, the performance still improves though is not as much as **Breakout (10x10x4)**. This is because conservative estimation can help avoid unsafe/suicidal actions, thereby the agent lives longer and gets high rewards. Additionally, As Reviewer nSvQ suggests, we add one more experiment on the full suite of Atari game **Pong (84x84x3)** and compare it with Rainbow in Appendix C.6 Figure 17. Our result shows that CEER achieves better performance and learns faster than Rainbow. These give us evidence that our method still works well in more complicated environments with continuous (or high-dimensional) state space.
> > >
> > > And, since we have shown a strong correlation between memory density and performance improvement, designing state abstraction methods to merge information from similar states is a significant direction, which can increase the memory density and further improve the performance.

---

> > > > ### Comment · Reviewer_nSvQ · 2022-11-21
> > > > **Thank you for the response**
> > > >
> > > > Thank you for the further clarification and the additional experiments. It is interesting and surprising that CEER can still work well in high-dimensional state space. This could also motivate future work. I recommend acceptance. I will change my score to Accept.

---

> > > > > ### Author Response · Authors · 2022-11-22
> > > > > **Thank you for your comments and feedback**
> > > > >
> > > > > We are glad that the reviewer's concerns have been addressed. We thank the reviewer nSvQ for the time and constructive feedback.

---

> ### Author Response · Authors · 2022-11-12
> **Response to Reviewer nSvQ, Part 2**
>
> > - In Eq. (3), the transition probabilities are estimated by visiting counts. In the case that a state has never been seen or an action has never been tried by the agent, how should the probabilities be estimated?
>
> We only consider transitions in the memory. If a state/action is not in the memory, it means there is no node/edge for that state/action, and the transition probability will be 0. This is our conservative approach, where we only use information that we have actually experienced.
>
> > - Can the methods be used in continuous state space? In this case, how to construct the graph, and how the value iteration can be applied?
>
> CEER builds a discrete graph to obtain its conservative estimation, which is not very suitable for continuous action domains. We mention some ideas for future work on the continuous action case in Section 6 Conclusion.
> Since our work has shown that regularizers based on conservative estimation can benefit learning, we can try other approaches such as offline RL methods [1] to obtain the conservative estimation without building a graph explicitly. For some environments with continuous action space which are numerically suitable, we can use discretization techniques [4][5] first and then construct RM-MDP with a graph structure. This may be a practical approach since complicated games such as DOTA [2] and StarCraft [3] can be discretized.
>
> [1] Levine, Sergey, et al. "Offline reinforcement learning: Tutorial, review, and perspectives on open problems." arXiv preprint arXiv:2005.01643 (2020).
>
> [2] Vinyals, Oriol, et al. "Grandmaster level in StarCraft II using multi-agent reinforcement learning." Nature 575.7782 (2019): 350-354.
>
> [3] Berner, Christopher, et al. "Dota 2 with large scale deep reinforcement learning." arXiv preprint arXiv:1912.06680 (2019).
>
> [4] Kotsiantis, Sotiris, and Dimitris Kanellopoulos. "Discretization techniques: A recent survey." GESTS International Transactions on Computer Science and Engineering 32.1 (2006): 47-58.
>
> [5] Jin, Jiarui, et al. "Graph-Enhanced Exploration for Goal-oriented Reinforcement Learning." International Conference on Learning Representations. 2022.
>
> > - The paper only reports sample efficiency but not wall-clock time. It is unclear how many additional computational resources are needed for the graph construction and value iteration.
>
> We show the training speed of each method in Appendix C.5 Table 4 with metric Frames Per Second (FPS).
> CEER can interact 308 frames per second with the environment, which is similar to PER (318), slightly slower than DQN (402), and much faster than DreamerV2 (19). This shows that the extra computation of our method is acceptable.
>
> > - How the memory density in Eq. (10) could be applied in continuous state space.
>
> Currently, the memory density for discrete action space is based on the number of actions. We mention some interesting future work for continuous action space in Section 5.4. For example, we can discretize the action space with a clustering method such as KNN, or we can use the policy variance at each state to measure the density.

---

### Official Review · Reviewer_MxYs · 2022-10-27

**Confidence:** 4
**Correctness:** 3
**Technical Novelty And Significance:** 3
**Empirical Novelty And Significance:** 3
**Recommendation:** 6

**Clarity, Quality, Novelty And Reproducibility:**

**Clarity:** As I understood the study easily, I believe the authors clearly explain the majority of the work.

**Quality:** This is a high-quality study. The approach is novel, in my opinion, and the empirical studies satisfy the deep RL benchmarking standards.

**Novelty:** The idea is very intuitive, and the proposed framework is novel.

**Reproducibility:** One of my main concerns is reproducibility. Although the experimental setup and implementation are broadly explained, there is no code given, even in the supplementary material. I strongly suggest the authors provide the source code for their project, either in a .zip file or an anonymous GitHub repository. Open an anonymous account, upload the code to a repo, and anonymize the repo to provide a link in the paper so that no one can identify the authors. This is a major concern.

**Strength And Weaknesses:**

**Strengths:**
- The paper is well-written and easy to follow.
- There are not many typos or punctuation mistakes as far as I noticed.
- The motivation is great, that is, considering the importance of neither temporally correlated nor single transitions but the entire replay buffer.
- Constructing an MDP over the collected transitions, and solving it through a modified dynamic programming scheme is novel.
- The authors perform a credible set of experiments on various benchmarks, such as using 20 random seeds and training for 2 million time steps are promising. The experimental setup and implementation details are explained in depth (e.g., hyper-parameter tuning).
- A comprehensive supplementary material is provided that enlights the depth of the study.

**Weaknesses:**
- The paper lacks a discussion over the competing methods. The authors only translate the results given in the Tables and Figures into sentences. Please describe in detail why competing methods exhibit poor performance, and what the reasons are. Focusing only on the proposed method in discussions decreases the explainability of the work and the introduced novelties.
- The authors solve an MDP over the created graph according to the transitions contained in the replay buffer. However, in such a case, the MDP is always changing as new transitions are added to the buffer. Hence, this MDP is non-stationary. The authors did not address this.
- The proposed method relies on prior offline/batch reinforcement learning approaches as the authors provide a reference to the corresponding papers. Moreover, the authors mention Experience Replay (ER) with reweighted updates, ER with episodic RL, and ER with reverse updates. The proposed framework contains similar entities from these three categories. I suggest the authors also discuss the most similar approach to their method in detail so that the novelty becomes more clear. Moreover, the work is very similar to the [paper](https://openreview.net/forum?id=HkxjqxBYDB) by Zhu et al. Please clarify this.
- The transition probabilities in Eq. (3) are calculated using visitation counts. How should the probabilities be calculated if the agent has never experienced a state or performed a certain action?
- The proposed method is _clearly for discrete action spaces_ as I believe constructing a graph for continuous action domains is impossible even under discretized environments because a slight numerical deviation may yield significant results in especially locomotion tasks and result in loss of information.
- Conservative regularization is used for Q-network and policy to make their update closer to the MDP corresponding to the collected transitions. However, if the behavioral policy never improves (e.g., choosing similar actions for similar states), how come the proposed method is expected to advance the underlying RL algorithm, as the updates are conservatively regularized? Specifically, the conservative updates prevent the RL algorithm from taking large gradient steps; hence, the policy may never improve. This would go on recursively. Am I correct? The authors should discuss this.
- Although the introduced performance metric, ARPI, seems to be a reasonable proxy for evaluating the performances, why didn't the authors use other metrics as given in the [work](https://ojs.aaai.org/index.php/AAAI/article/view/11694) of Henderson et al.? At least the authors could've discussed the existing metrics given in that [study](https://ojs.aaai.org/index.php/AAAI/article/view/11694) and maybe tried one of them.

**Detailed Comments**:
- Although the related work gives insights into the current approaches to experience replay, I believe the following work is worth discussing as they focus on either the Prioritized Experience Replay algorithm or reweighting the importance of transitions:
1) [An Equivalence between Loss Functions and Non-Uniform Sampling in Experience Replay](https://papers.nips.cc/paper/2020/hash/a3bf6e4db673b6449c2f7d13ee6ec9c0-Abstract.html) by Fujimoto et al.
2) [Actor Prioritized Experience Replay](https://arxiv.org/abs/2209.00532) by Saglam et al.
3) [Model-augmented Prioritized Experience Replay](https://openreview.net/forum?id=WuEiafqdy9H) by Oh et al.
4) [Off-Policy Correction for Deep Deterministic Policy Gradient Algorithms via Batch Prioritized Experience Replay](https://ieeexplore.ieee.org/abstract/document/9643162) by Cicek et al.
5) [Experience Replay Optimization](https://www.ijcai.org/proceedings/2019/589) by Zha et al.
- Fifth line in Section 3: Replace "expected cumulative rewards" with "expected discounted sum of rewards"
- First line on page 3: Give a reference to the [Dynamic Programming book](https://www.degruyter.com/document/doi/10.1515/9781400835386/html) when mentioning Bellman optimality equation
- Third line on page 3: No need to give a reference when mentioning deep neural networks
- Final paragraph in Section 3: There are sentences: _"Though the replay memory in online learning
is usually a time-changing first in first out queue, at a specific time step, the memory can be
considered as a static dataset that contains finite transitions. Thus, solving RM-MDP is similar to
offline RL setting that solves an empirical MDP and finally results in a conservative estimate."_ Are there any references?
- Fourth sentence in Section 4.3: There is a sentence: _"Our ˆQ value from RM-MDP is constrained in the replay memory that will never overestimate."_ The overestimation is induced by the maximization in the Q-network updates. Similarly, the proposed method also uses maximization even if across the recorded transitions. This sentence should be explained in more detail or a reference should be given, as it is a very strong claim.
- Please express the considered baseline algorithms in the full form, not using acronyms.

**Summary Of The Paper:**

By considering it as an empirical Replay Memory MDP (RM-MDP), the authors in this study take advantage of the information contained in the experience replay memory. The authors discovered a conservative value estimate that solely considers transitions seen within the replay memory by solving it using dynamic programming. Based on this conservative estimate, value and policy regularizers are developed and integrated with model-free learning algorithms. To gauge the effectiveness of RM-MDP, they develop a memory density metric. According to the authors' empirical findings, there is a significant correlation between memory density and performance enhancement. They use a technique called Conservative Estimation with Experience Replay (CEER), which significantly boosts sampling efficiency, mainly when the memory density is high. Such a conservative approximation can nonetheless assist in preventing suicidal behavior and hence increase performance even when the memory density is low.

**Summary Of The Review:**

Although the paper combines model-based and model-free methods, many other papers have done the same. There hasn't been any discussion of their connections. Additionally, it is clear that the strategy could not be scalable to continuous state space. Moreover, the reproducibility of the introduced work is a major concern.

---

> ### Author Response · Authors · 2022-11-12
> **Response to Reviewer MxYs, Part 1**
>
> Thank you for the review, and for providing us with thorough questions and suggestions! We discuss these points in detail below. Please let us know if there are further concerns.
>
> > - The paper lacks a discussion over the competing methods. The authors only translate the results given in the Tables and Figures into sentences. Please describe in detail why competing methods exhibit poor performance, and what the reasons are. Focusing only on the proposed method in discussions decreases the explainability of the work and the introduced novelties.
>
> We added more discussion of baseline methods in Section 5.3 Overall Performance as follows. We hope this will give more insights why our method performs better.
>
> For these baseline methods, PER samples transitions with high TD error more frequently. However, the TD error changes dynamically as the network is updated, which may hurt the performance [1].
> EBU and TER consider the sequential character of trajectories and sample transitions in reversed order.
> Our method propagates rewards when solving the RM-MDP, and thereby also considers the character of trajectories. Furthermore, our regularizers update the network in a supervised manner, which is more stable than TD learning.
> DisCor samples transitions proportional to the accuracy of the target value, but such transitions are not necessarily the most related to the optimal policy.
> In contrast, our method aims to provide accurate target values by combining the conservative estimation $\hat{Q}$ from RM-MDP with the network estimation $Q$, and treats all transitions as equally important. This can provide accurate target values while not missing the most related transitions.
> In summary, our method combines the advantages of several previous methods, and our experimental results indicate that it is suitable for a large variety of different tasks.
>
> [1] Cicek, Dogan C., et al. "Off-Policy Correction for Deep Deterministic Policy Gradient Algorithms via Batch Prioritized Experience Replay." 2021 IEEE 33rd International Conference on Tools with Artificial Intelligence (ICTAI). IEEE, 2021.
>
> > - The authors solve an MDP over the created graph according to the transitions contained in the replay buffer. However, in such a case, the MDP is always changing as new transitions are added to the buffer. Hence, this MDP is non-stationary. The authors did not address this.
>
> Yes, during the interaction with the environment, RM-MDP is non-stationary.
> Since each step only adds one new transition to the memory, we update $\hat{Q}$ in a sampling manner. This is a trade-off between estimation accuracy and computation burden. We add one more experiment in Appendix C.8 to show that the sampling update is enough to provide an accurate estimation. We compare the performance of using Sampling Update and using Sweeping Update. Sweeping Update updates all state-action pairs at every step, which address the non-stationary problem. From the results in Figure 18, we find no difference in performance, which means Sampling Update is enough to provide an accurate estimation.
>
> Another benefit is the training speed. Table 4 in Appendix C.5 shows that with this Sampling Update, CEER can interact 308 frames per second (FPS) with the environment, which is similar to PER (318) and slightly slower than DQN (402). This highlights that besides improving performance, the extra computation cost of our method is acceptable.

---

> ### Author Response · Authors · 2022-11-12
> **Response to Reviewer MxYs, Part 2**
>
> > - The proposed method relies on prior offline/batch reinforcement learning approaches as the authors provide a reference to the corresponding papers. Moreover, the authors mention Experience Replay (ER) with reweighted updates, ER with episodic RL, and ER with reverse updates. The proposed framework contains similar entities from these three categories. I suggest the authors also discuss the most similar approach to their method in detail so that the novelty becomes more clear. Moreover, the work is very similar to the paper by Zhu et al. Please clarify this.
>
> Thank you for your suggestion. We revised Section 2 Related Work to address the novelty in our contribution. We explain the difference between our work and Zhu et al. (ERLAM) [1] as follows:
>
> Besides the regularizers we designed, there are other differences and contributions in our work:
> ERLAM maintains a second graph memory to store good past experiences, and updates the values of the graph in a reversed breadth-first manner. This way is not suitable for stochastic environments, and requires techniques to handle cycles in the graph.
> In contrast, our method directly builds a graph-based replay memory, which constructs an empirical MDP over the collected transitions. We then solve this MDP through a modified dynamic programming scheme.
> Our method is therefore more general and has no constraints on the environment MDP. For example, it can be stochastic and can have cycles.
>
> Additionally, from our experiments, we find a strong correlation between memory density and performance improvement. This relationship gives us a clear understanding of the circumstances when our method will help most. This strong relationship definitely can guide future work to diagnose and design new methods.
>
> In the high-level view, our method focuses on conservative estimation from replay memory. Constructing a graph is a way to obtain such a conservative estimate, but it is not the only way. There are many offline RL methods [2] for solving the empirical MDP which result in a conservative estimate. Our method shows a path for learning from data in an online replay memory by using offline static dataset methods, which we think is a promising direction for further future work.
>
> [1] Zhu, Guangxiang, et al. "Episodic Reinforcement Learning with Associative Memory." International Conference on Learning Representations. 2019.
>
> [2] Levine, Sergey, et al. "Offline reinforcement learning: Tutorial, review, and perspectives on open problems." arXiv preprint arXiv:2005.01643 (2020).
>
> > - The transition probabilities in Eq. (3) are calculated using visitation counts. How should the probabilities be calculated if the agent has never experienced a state or performed a certain action?
>
> We only consider transitions in the memory. If a state/action is not in the memory, it means there is no node/edge for that state/action, and the transition probability will be 0. This is our conservative approach, where we only use information that we have actually experienced.
>
> > - The proposed method is clearly for discrete action spaces as I believe constructing a graph for continuous action domains is impossible even under discretized environments because a slight numerical deviation may yield significant results in especially locomotion tasks and result in loss of information.
>
> CEER builds a discrete graph to obtain its conservative estimation, which is not very suitable for continuous action domains. We mention some ideas for future work on the continuous action case in Section 6 Conclusion.
> Since our work has shown that regularizers based on conservative estimation can benefit learning, we can try other approaches such as offline RL methods [1] to obtain the conservative estimation without building a graph explicitly. For some environments with continuous action space which are numerically suitable, we can use discretization techniques [4][5] first and then construct RM-MDP with a graph structure. This may be a practical approach since complicated games such as DOTA [2] and StarCraft [3] can be discretized.
>
> [1] Levine, Sergey, et al. "Offline reinforcement learning: Tutorial, review, and perspectives on open problems." arXiv preprint arXiv:2005.01643 (2020).
>
> [2] Vinyals, Oriol, et al. "Grandmaster level in StarCraft II using multi-agent reinforcement learning." Nature 575.7782 (2019): 350-354.
>
> [3] Berner, Christopher, et al. "Dota 2 with large scale deep reinforcement learning." arXiv preprint arXiv:1912.06680 (2019).
>
> [4] Kotsiantis, Sotiris, and Dimitris Kanellopoulos. "Discretization techniques: A recent survey." GESTS International Transactions on Computer Science and Engineering 32.1 (2006): 47-58.
>
> [5] Jin, Jiarui, et al. "Graph-Enhanced Exploration for Goal-oriented Reinforcement Learning." International Conference on Learning Representations. 2022.

---

> ### Author Response · Authors · 2022-11-12
> **Response to Reviewer MxYs, Part 3**
>
> > - Conservative regularization is used for Q-network and policy to make their update closer to the MDP corresponding to the collected transitions. However, if the behavioral policy never improves (e.g., choosing similar actions for similar states), how come the proposed method is expected to advance the underlying RL algorithm, as the updates are conservatively regularized? Specifically, the conservative updates prevent the RL algorithm from taking large gradient steps; hence, the policy may never improve. This would go on recursively. Am I correct? The authors should discuss this.
>
> Thank you for this interesting question. DQN uses $\epsilon$-greedy ($\epsilon:  1 \rightarrow 0.1$) to trade off exploration and exploitation. Though we regularize the $Q$ network with conservative estimation, the $\epsilon$ greedy approach will try unknown actions and thus the behavioral policy will improve.
>
> Furthermore, for the greedy selection due to the $Q$ network, since DQN suffers from overestimation, the combination with conservative estimation can help get a more accurate target value when updated and shift away from being stuck on bad actions.
>
> Another potential benefit is similar to Go-Explore [1][2], which returns to promising states first before intentionally exploring. The conservative regularizers in CEER help the agent to first go to the frontier of unfamiliar states from familiar states, and then collect new transitions and explore from the frontier. This helps achieve a thorough exploration of the environment and can avoid unnecessary suicidal actions.
>
> [1] Ecoffet, Adrien, et al. "Go-explore: a new approach for hard-exploration problems." arXiv preprint arXiv:1901.10995 (2019).
>
> [2] Ecoffet, A., Huizinga, J., Lehman, J. et al. First return, then explore. Nature 590, 580–586 (2021). https://doi.org/10.1038/s41586-020-03157-9
>
> > - Although the introduced performance metric, ARPI, seems to be a reasonable proxy for evaluating the performances, why didn't the authors use other metrics as given in the work of Henderson et al.? At least the authors could've discussed the existing metrics given in that study and maybe tried one of them.
>
> Yes, we used several of the metrics discussed in Henderson et al. [1] to report our results. For examples, we run 20 seeds for each method; we report the mean value rather than the top-$N$ trials; we record standard error, which is a similar metric to confidence interval.
> However, we still need a metric to measure the performance improvement for the whole learning process which is missing in [1]. That is the main reason for introducing a new metric.
>
> [1] Henderson, Peter, et al. "Deep reinforcement learning that matters." Proceedings of the AAAI conference on artificial intelligence. Vol. 32. No. 1. 2018.
>
> > - Although the related work gives insights into the current approaches to experience replay, I believe the following work is worth discussing as they focus on either the Prioritized Experience Replay algorithm or reweighting the importance of transitions:
>
> Thank you for your suggestion, we added these in Section 2 Related Work.
>
> > - Fourth sentence in Section 4.3: There is a sentence: "Our ˆQ value from RM-MDP is constrained in the replay memory that will never overestimate." The overestimation is induced by the maximization in the Q-network updates. Similarly, the proposed method also uses maximization even if across the recorded transitions. This sentence should be explained in more detail or a reference should be given, as it is a very strong claim.
>
> We apologize for the inaccurate description. When combining $Q$-learning with function approximation, DQN suffers from substantial overestimation in some games [1]. Since $\hat{Q}$ from the RM-MDP is a conservative estimation and is estimated in a tabular manner without function approximation, it is not so prone to overestimation. We rewrite the description in the new version in Section 4.3.
>
> [1] Van Hasselt, Hado, Arthur Guez, and David Silver. "Deep reinforcement learning with double q-learning." Proceedings of the AAAI conference on artificial intelligence. Vol. 30. No. 1. 2016.
>
> > - One of my main concerns is reproducibility.
>
> Thank you so much for your interest in our code. This is the [link](https://drive.google.com/file/d/1n__AbMyHmKcZkjGg-_HiG5HsVAT9yTED/view?usp=share_link
> ) for our code.

---

> > ### Comment · Reviewer_MxYs · 2022-11-21
> > **Thank you for the clarifications**
> >
> > I would like to thank the authors for their responses and efforts. It is nice to highlight the changes with the color red.
> >
> > ### Response from the reviewers
> > -**Conservative regularization for the Q-network(s):** The use of $\epsilon$-greedy can indeed explore the outcomes of untried actions. But please include your explanation in the revised manuscript. The use of a sophisticated exploration algorithm can be a part of promising future work.
> >
> > -**Performance metric:** OK, as far as I'm concerned.
> >
> > -**Section 2 Related Work:** Thank you for addressing my concern.
> >
> > -**Overestimation of the Q-network when used with conservative estimation:** OK, as far as I'm concerned.
> >
> > -**Reproducibility:** I reached the code, and it's OK.
> >
> > -**Conclusion:** Other than these, I still have the following concerns.
> >
> >
> >
> > ### Current issues
> > I couldn't find any response to my previous comments, given below. Maybe I missed it, but please reply promptly so that I can increase my score.
> >
> > _The transition probabilities in Eq. (3) are calculated using visitation counts. How should the probabilities be calculated if the agent has never experienced a state or performed a certain action?_
> >
> > _The proposed method is clearly for discrete action spaces as I believe constructing a graph for continuous action domains is impossible even under discretized environments because a slight numerical deviation may yield significant results in especially locomotion tasks and result in loss of information._

---

> > > ### Author Response · Authors · 2022-11-21
> > > **Response to Reviewer MxYs's further issues**
> > >
> > > We thank reviewer MxYs for your positive support. We will add the explanation of conservative regularization for the Q-network(s) in the next version. We explain the two issues below and are sorry about the misordering of the three parts of the responses.
> > >
> > > > - The transition probabilities in Eq. (3) are calculated using visitation counts. How should the probabilities be calculated if the agent has never experienced a state or performed a certain action?
> > >
> > > We only consider transitions in the memory. If a state/action is not in the memory, it means there is no node/edge for that state/action, and the transition probability will be 0. This is our conservative approach, where we only use information that we have actually experienced.
> > >
> > > > - The proposed method is clearly for discrete action spaces as I believe constructing a graph for continuous action domains is impossible even under discretized environments because a slight numerical deviation may yield significant results in especially locomotion tasks and result in loss of information.
> > >
> > > CEER builds a discrete graph to obtain its conservative estimation, which is not very suitable for continuous action domains. We mention some ideas for future work on the continuous action case in Section 6 Conclusion.
> > > Since our work has shown that regularizers based on conservative estimation can benefit learning, we can try other approaches such as offline RL methods [1] to obtain the conservative estimation without building a graph explicitly. For some environments with continuous action space which are numerically suitable, we can use discretization techniques [4][5] first and then construct RM-MDP with a graph structure. This may be a practical approach since complicated games such as DOTA [2] and StarCraft [3] can be discretized.
> > >
> > > [1] Levine, Sergey, et al. "Offline reinforcement learning: Tutorial, review, and perspectives on open problems." arXiv preprint arXiv:2005.01643 (2020).
> > >
> > > [2] Vinyals, Oriol, et al. "Grandmaster level in StarCraft II using multi-agent reinforcement learning." Nature 575.7782 (2019): 350-354.
> > >
> > > [3] Berner, Christopher, et al. "Dota 2 with large scale deep reinforcement learning." arXiv preprint arXiv:1912.06680 (2019).
> > >
> > > [4] Kotsiantis, Sotiris, and Dimitris Kanellopoulos. "Discretization techniques: A recent survey." GESTS International Transactions on Computer Science and Engineering 32.1 (2006): 47-58.
> > >
> > > [5] Jin, Jiarui, et al. "Graph-Enhanced Exploration for Goal-oriented Reinforcement Learning." International Conference on Learning Representations. 2022.

---

> > > > ### Comment · Reviewer_MxYs · 2022-11-22
> > > > **I've increased my score**
> > > >
> > > > I'd like to thank the authors for their quick response. I'm fully satisfied with the work, its contributions, and the authors' justifications. I've increased my score.
> > > >
> > > > There is a promising direction for future work per the authors. I believe they should be considered.

---

> > > > > ### Author Response · Authors · 2022-11-22
> > > > > **Thank you for your comments and feedback**
> > > > >
> > > > > We are glad that the reviewer's concerns have been addressed. We thank the reviewer MxYs for the time and constructive feedback.

---

### Official Review · Reviewer_PqTv · 2022-10-30

**Confidence:** 4
**Correctness:** 4
**Technical Novelty And Significance:** 3
**Empirical Novelty And Significance:** 3
**Recommendation:** 6

**Clarity, Quality, Novelty And Reproducibility:**

I like this paper in general. The motivation of this paper is clear and make senses to me. The clarity is good. Related work is fully survey. The authors discuss about both limitation and advantages of CEER.

The results are significant, and the writing quality is good. The proposed approach is novel. The authors report most of hyper-paremeter settings. I believe this approach can be reproduced.

Typos:
1. Section 5.3, outperforms other baselines to a large margin -> by a large margin

**Strength And Weaknesses:**

Strength:
+ The perspective of treating the replay memory as an empirical MDP is interesting, and the proposed algorithms successfully improve sample efficiency by leveraging this empirical MDP.
+ The authors conduct a lot of solid experimental analysis to demonstrate the effectiveness of the methods, and provide interesting insights for future direction based on this work.

Weaknesses:
- The proposed method is only evaluated on limited environments. I think experiments on more complicated environments such as full suite of atari games are necessary to make the work more solid and convincing.
- The baseline considered in this work is not strong enough. It would be better to see if the CEER techniques can improve sample efficiency upon more powerful existing RL algorithms such as RAINBOW.

**Summary Of The Paper:**

This paper proposes a new way to make experience replay more efficient. Inspired by episodic learning, the authors treat the replay memory as an empirical replay memory MDP (RM-MDP). With dynamic programming, conservative value esimtate is learned by only considering transitions observed in the replay memory. Based on the empirical estimated dynamics, the authors propose value regularizer and policy regularizer and use these two regularizers to boost the learning process of DQN. To better understand the proposed method, the authors design metric for memory density, and empirically find a strong correlation between performance improvement and memory density. Extensive experiments show that the proposed CEER algorithm improves sample efficiency across several benchmarks in discrete action domains.

**Summary Of The Review:**

The paper proposes a new way to make replay memory more efficient. The authors conduct thorough empirical analysis to demonstrate the advantages of this work. I think this work could benefit future research on replay memory in RL community. However, further experiments on more complicated should be demonstrated to make the proposed algorithms more convincing.

---

> ### Author Response · Authors · 2022-11-12
> **Response to Reviewer PqTv**
>
> Thank you for the review! We are glad that you like our work. We answer the questions below.
>
> > - The proposed method is only evaluated on limited environments. I think experiments on more complicated environments such as full suite of atari games are necessary to make the work more solid and convincing.
> > - The baseline considered in this work is not strong enough. It would be better to see if the CEER techniques can improve sample efficiency upon more powerful existing RL algorithms such as RAINBOW.
>
> We add one more experiment on the Atari game Pong and compare with Rainbow in Appendix C.6 Figure 17. Our result shows that CEER achieves better performance and learns faster than Rainbow. This gives us evidence that our method still works well on more complicated environments. In the future, we can do experiments on more complicated environments to explore the limits of our method.
>
> As Reviewer nSvQ suggests, we provide additional results that compare with further model-based baselines Dyna [1], MVE [2], MCTS [3], and DreamerV2 [4] in Appendix C.4 Figure 15, 16, to show that our method is still competitive. Though our work is not a model-based method in the strict sense, we hope these experiments can provide a more comprehensive comparison and make our evaluation more solid.
>
> [1] Sutton, Richard S. "Dyna, an integrated architecture for learning, planning, and reacting." ACM Sigart Bulletin 2.4 (1991): 160-163.
>
> [2] Feinberg, Vladimir, et al. "Model-based value estimation for efficient model-free reinforcement learning." arXiv preprint arXiv:1803.00101 (2018).
>
> [3] Hamrick, Jessica B., et al. "Combining Q-Learning and Search with Amortized Value Estimates." International Conference on Learning Representations. 2019.
>
> [4] Hafner, Danijar, et al. "Mastering Atari with Discrete World Models." International Conference on Learning Representations. 2020.

---

### Decision · Program_Chairs · 2023-01-20

**Decision:**

Accept: poster

**Justification For Why Not Higher Score:**

The method makes intuitively a lot of sense but does not come with any formal guarantees While working well empirically with continuous/high-dimensional states combined with hashing, but I still have doubts how general this is.

**Justification For Why Not Lower Score:**

All reviewers recommend accepting the paper.

**Metareview: Summary, Strengths And Weaknesses:**

Summary:
The paper proposes to learn a graph-based MDP approximation from the data in the replay buffer, which is then solved and used to regularize both the policy and value function. They also develop a metric that can indicate whether the approach will work well for a given replay buffer. The approach is validated experimentally

Strengths:
- Well written paper
- Innovative approach
- Most questions resolved in discussion

Weaknesses:
- The approach is designed based on discrete states and discrete actions. The experiments show empirically that it also works well for large and continuous state spaces, there are some ideas on how to deal with continuous action spaces.
- The discussion on why baselines perform better has improved a lot, but still reads a bit more like hypotheses

**Note From Pc:**

if the above contains the word "oral" or "spotlight" please see: "oral" presentation means -> notable-top-5% and "spotlight" means -> notable-top-25%. As stated in our emails, we are disassociating presentation type from AC recommendations

**Summary Of Ac-Reviewer Meeting:**

N/A